# Progressive coevolution of the yeast centromere and kinetochore

Jana Helsen[1,2 ✉], Kausthubh Ramachandran[2,3], Gavin Sherlock[1 ✉] & Gautam Dey[2 ✉]

During mitosis, stable but dynamic interactions between centromere DNA and the kinetochore complex enable accurate and efficient chromosome segregation. Even though many proteins of the kinetochore are highly conserved[1,2], centromeres are among the fastest evolving regions in a genome[3,4], showing extensive variation even on short evolutionary timescales. Here we sought to understand how organisms evolve completely new sets of centromeres that still effectively engage with the kinetochore machinery by identifying and tracking thousands of centromeres across two major fungal clades, including more than 2,500 natural strain isolates and representing over 1,000 million years of evolution. We show that new centromeres spread progressively via drift and subsequent selection and that the kinetochore, which is evolving slowly in relative terms, appears to act as a filter to determine which new centromere variants are tolerated. Together, our findings provide insight into the evolutionary constraints and trajectories shaping centromere evolution.

Centromeres are indispensable chromosomal elements for cell division across eukaryotes. As attachment points for the chromosome segregation machinery, they are responsible for partitioning cellular DNA rapidly and reproducibly through a stable but dynamic interaction with the kinetochore complex and spindle microtubules (Fig. 1a). Although each centromere needs to accomplish this essential cellular feat alongside a conserved set of kinetochore proteins[1,2], centromeres are among the fastest evolving regions in the genome[3,4], showing striking variability across the tree of life[5,6] (Fig. 1b and Supplementary Data 1). Centromeres range from complex epigenetically defined regions of hundreds of kilobases embedded in megabase-sized arrays of satellite DNA (regional centromeres) in metazoans[7] and plants[8] to genetically defined loci (point centromeres) of 100–200 nucleotides in budding yeasts[9–11].

Unlike protein-coding genes, which are often only present as single copies in the genome, species with monocentric chromosomes possess multiple centromeres, one on each chromosome. For organisms to evolve centromeres with new features, this means that each chromosome must either alter its existing centromere or acquire a new centromere (Fig. 1c), making centromere evolution conceptually very different from gene evolution. To date, many of the basic evolutionary principles underlying centromere evolution remain largely unknown. It remains unclear (1) whether centromere transitions occur progressively or concurrently; (2) how much they are a result of selection and/or drift; and (3) how these rapidly evolving regions ensure that their crucial connections with the kinetochore and spindle are maintained. It has been particularly challenging to address these questions without the fine-grained genomic data and high-throughput centromere detection algorithms necessary to track regional centromeres through large swathes of evolutionary time. In this study, we use a combination of centromere discovery, phylogenetic profiling and in vivo functional assays to determine the evolutionary constraints and trajectories of centromere transitions in two major fungal clades. We first rigorously test models of centromere evolution using the point centromeres of budding yeasts and then extend our approach to the more complex centromeres of the Mucoromycota[12].

## Mapping centromere sequence landscapes

To systematically explore the evolutionary mechanisms driving centromere transitions, centromere diversity needs to be mapped across the phylogenetic tree in the context of recent evolution. We selected the Saccharomycetaceae, a fungal clade including the model species *Saccharomyces cerevisiae* and *Nakaseomyces glabratus* (formerly *Candida glabrata*) to identify and characterize such transitions, as they have short, genetically defined point centromeres[13] and use a structurally similar spindle to segregate their chromosomes[14,15] and the clade encompasses many species in a relatively short evolutionary time frame[16]. Each point centromere is defined by an AT-rich region (centromere DNA sequence element II (CDEII) and two DNA motifs (CDEI and CDEIII)[17] (Fig. 1d) that are bound by specific DNA-binding proteins, several of which are unique to the clade[18]. Although it is known that point centromeres vary across the clade, sequences are only available for a handful of species[9,18], insufficient to pinpoint when and how centromere transitions occurred. To compile a systematic list of centromere sequences across the Saccharomycetaceae clade, we developed an automated point centromere annotation tool (PCAn)[19] that uses two sequential motif searches to detect point centromeres in genome assemblies (Fig. 1e) (see the Methods for details).

Using PCAn to annotate centromeres for 138 species, we generated a comprehensive atlas of point centromere diversity across approximately 114 million years (Myr) of evolution (Fig. 1f, Extended Data Figs. 1 and 2 and Supplementary Data 2). The numbers of predicted

[1]Department of Genetics, Stanford University School of Medicine, Stanford, CA, USA. [2]Cell Biology and Biophysics, European Molecular Biology Laboratory, Heidelberg, Germany. [3]Collaboration for joint PhD degree between EMBL and Heidelberg University, Faculty of Biosciences, Heidelberg University, Heidelberg, Germany. ✉e-mail: jana.helsen@embl.de; gsherloc@stanford.edu; gautam.dey@embl.de

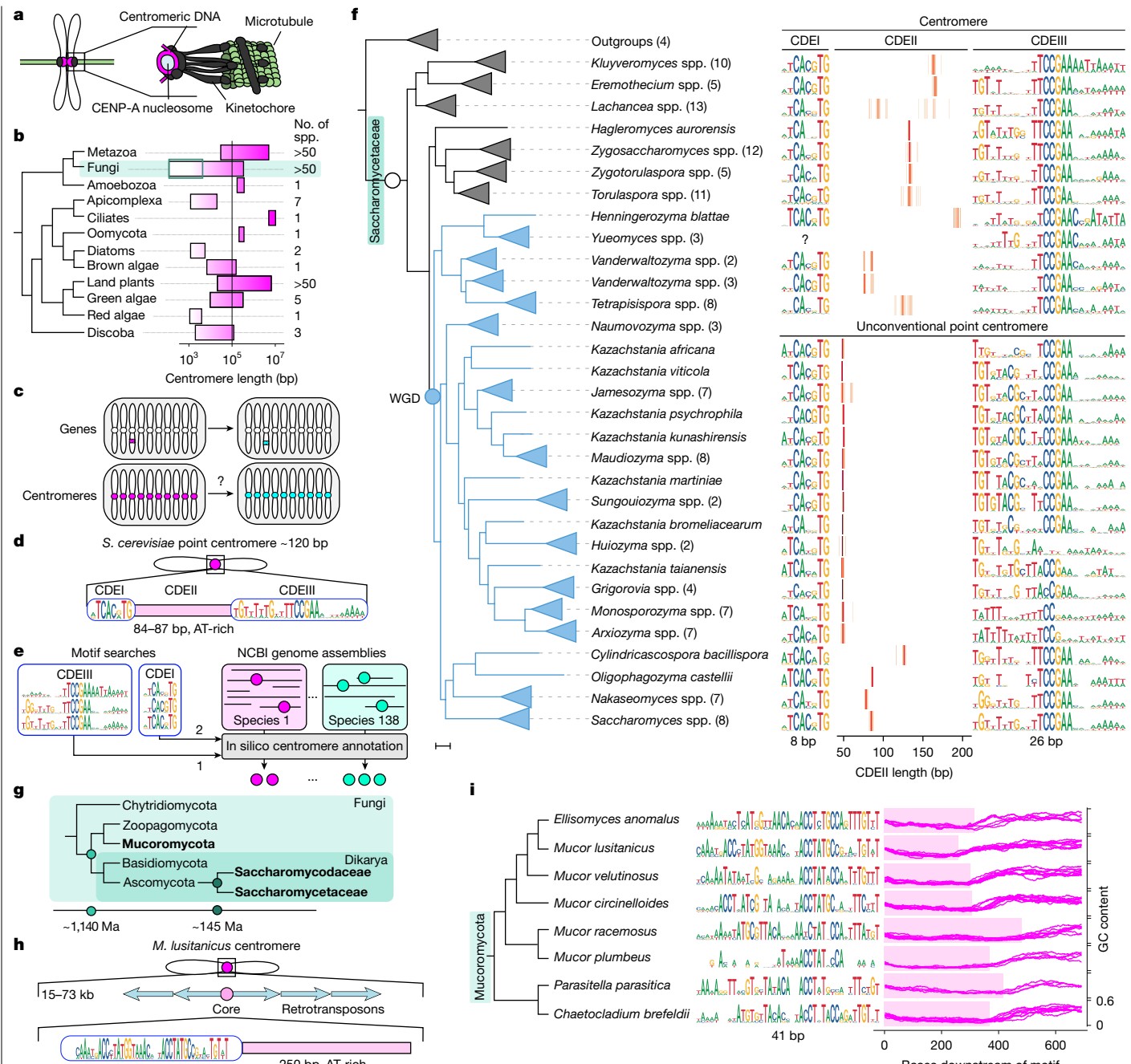

**Fig. 1 | The sequence landscape of motif-defined centromeres. a**, Stylized schematic of centromeric DNA bound to a microtubule through the kinetochore complex[44]. **b**, Minimum and maximum sizes of known and predicted centromeres across eukaryotes. The number of species with known centromeres is indicated on the right for each clade. The green box represents the approximate size range of the centromeres studied here. **c**, Schematic highlighting the complexity of centromere evolution. **d**, Structure of the *S. cerevisiae* point centromere. **e**, Simplified schematic representing the in silico PCAn pipeline. **f**, Landscape of predicted centromere sequences across Saccharomycetaceae. Species phylogeny was determined using a concatenation-based maximum-likelihood (ML) analysis of 1,270 orthologous groups of proteins using the Le and Gascuel model with four rate categories (LG + G4). Genera are represented by triangles with the number of species in parentheses. Branches of species that emerged after the whole-genome duplication (WGD) are coloured in blue.

Outgroup species: *Wickerhamomyces anomalus*, *Candida albicans*, *Pichia kudriavzevii* and *Yarrowia lipolytica*. Centromere sequences are represented by DNA logos for CDEI and CDEIII, with graphs indicating CDEII length in between. *Naumovozyma* spp. have unconventional point centromeres lacking regular CDEI and CDEIII motifs[45]. Centromere profiles and predictions for *Naumovozyma* spp. can be found in Extended Data Fig. 2a–c and Supplementary Data 2. Tree scale, 0.1. **g**, Simplified fungal tree highlighting the clades in which we looked for and found motif-defined centromeres (bold type), with estimated divergence times in million years ago (Ma)[22,46]. **h**, The structure of the mosaic *M. lusitanicus* centromere[12]. **i**, Predicted centromeres for eight Mucoromycota species. Centromere sequences are represented by DNA logos for the motif, followed by graphs indicating the GC content of the region downstream of the motif. The approximate length of the AT-rich region is indicated by pink boxes. Schematic in **a** adapted with permission from ref. 44, Elsevier.

centromeres correspond well with published chromosome numbers (Extended Data Fig. 3), validating our approach while highlighting that our tool can also be used to obtain karyotype information. Despite

the limited diversity in inner kinetochore composition, point centromeres show extensive diversity across the clade. In some lineages, parts of the CDEIII motif are either completely conserved or eroded

away (for example, *Huiozyma* spp. lack the conserved CCGAA motif, whereas part of the CDEIII motif is more conserved in *Sungouiozyma* spp.). However, perhaps the most striking variation can be observed in the length of CDEII, the AT-rich region that wraps around the centromeric nucleosome[20,21]. CDEII length ranges from around 50 bp in many *Kazachstania*-related species to nearly 200 bp in *Henningerozyma blattae*. Notably, although there is extensive CDEII length variation between species, the length distribution in any given genome is remarkably consistent (Extended Data Fig. 1). Our atlas of centromere diversity shows that point centromeres have evolved extensively since their origination and that centromere transitions occurred multiple times in the Saccharomycetaceae clade.

PCAn also accurately detects centromeres right outside of the Saccharomycetaceae, identifying some of the recently described proto-point centromeres in a handful of Saccharomycodaceae[10] (Fig. 1g and Extended Data Fig. 2d). To determine whether our approach can also be used to identify key centromere features of more complex centromeres, we adapted our pipeline to detect the core kinetochore-binding region in Mucoromycota centromeres (Fig. 1g,h). Using the features previously identified in *Mucor lusitanicus*[12], we annotated core centromere regions for seven additional species (Fig. 1i and Supplementary Data 3). Remarkably, despite being separated from the Saccharomycetaceae by approximately 1,140 Myr (ref. 22) and lacking sequence homology, the core centromeres of *Mucor* also contain an AT-rich region that follows a similar pattern of length variation. In a single *Mucor* genome, the length of this region is remarkably consistent, yet it varies by nearly twofold between different species. Together, these findings demonstrate that our approach is widely applicable, capable of identifying both recent transitions in key centromere features and more universal evolutionary patterns.

## Centromeres evolve progressively

Next, we used our comprehensive atlas of centromere diversity to explore the evolutionary dynamics underlying centromere transitions. Given the substantial interspecific variation yet limited intraspecific variability in CDEII length, we focused on the evolutionary trajectories of this key centromere feature. Full transitions of CDEII length occurred independently on multiple occasions (Fig. 2a–c), even in the same genus (Fig. 2b,c). Unexpectedly, we also found some species carrying CDEII of two distinct lengths simultaneously, that is, species in which two distinct centromere types coexist in the same genome (Fig. 2c,d). This suggests that these species are currently in a state of centromere transition. In the *Jamesozyma* genus, although most species retain the ancestral CDEII of about 50 bp, *J. spencerorum* has undergone a complete transition to an approximately 60-bp CDEII, whereas its sister species *J. jinghongensis* has a nearly equal distribution of both centromere types (Fig. 2c and Extended Data Fig. 4a). Another example of mixed centromere states is found in the *Vanderwaltozyma* genus. All species in this genus possess a mixture of centromeres with a CDEII of either about 75 bp or about 85 bp, with the relative frequency of each variant varying across the clade. Although some species exhibit an equal distribution of both centromere types, others are enriched for either the shorter or longer variant (Fig. 2d). Similar patterns in length variation are found for the AT-rich regions of two Mucoromycota species (Fig. 2e), showing that this phenomenon is not unique to the budding yeast clade. These observations indicate that centromere transitions occur progressively: centromeres probably change one by one, with a transitional phase characterized by the coexistence of old and new centromere variants in the same genome.

Next, we sought to determine the role of selection in these progressive transitions. If different centromere variants can coexist in one cell, this implies that both can establish a sufficiently stable connection with the mitotic machinery. However, some variants might still be better at segregating and will thus be favoured by selection. To determine whether selection is required to achieve full centromere transitions, we conducted in silico centromere transition experiments (Fig. 2f,g and Extended Data Fig. 4b–d). Starting with an individual carrying 16 A type centromeres, the simulation iteratively transitions a single, randomly selected centromere to type B or not, on the basis of a retention probability. These simulations show that full centromere transitions are indeed not possible without selection for the new variant, both in haploids and diploids (Fig. 2f,g and Extended Data Fig. 4c). Even in a scenario in which the ancestral and new variants are equally good at segregating (drift alone causing frequency changes), full transitions are impossible. Similarly, selection for ancestral centromeres is required to maintain a full complement of ancestral variants (Fig. 2f and Extended Data Fig. 4b,d). In the examples shown in Fig. 2a–d, such full transitions took between about 23 and 84 Myr of evolution (Fig. 2h). To test centromere function in vivo and determine whether different centromere variants do indeed show differences in segregation efficiency before and after full transitions, we set up plasmid retention assays to determine the retention rate of plasmids carrying different types of centromeres (Fig. 2i and Extended Data Fig. 4e). Among the three *Jamesozyma* species we tested, only *J. spencerorum*, the species that underwent a complete transition to centromeres with a longer CDEII, had a strong preference for the longer variant (Fig. 2j), indicating that selection was indeed required for centromere transition in this clade. We hypothesize that centromere transitions occur progressively through a combination of selection and drift (Fig. 2k): differences in segregation efficiency determine which variants will be favoured by selection and which new variants are neutral and can spread through drift.

## New centromeres can spread through sex

Although the data shown above demonstrate the progressive nature of centromere transitions, they do not inform us about the mechanisms underlying the initial emergence and subsequent spread of new centromere variants in populations. To address this, we examined intraspecific variation. Using PCAn to annotate centromeres in 1,493 unique *S. cerevisiae* strains revealed that, although the majority contain 16 centromeres with consistent CDEII lengths (about 85 bp), approximately 9.5% harbour one or two variants with deviating CDEII lengths (<80 bp or >90 bp) (Fig. 3a, Extended Data Fig. 5a and Supplementary Data 4). To determine whether these 142 strains all carry the same variant centromere or 142 different ones and thus determine how many times independent variants arose and spread in the population, we assigned synteny and aligned centromere sequences and were able to assign 13 distinct variant centromeres in the *S. cerevisiae* population (Fig. 3b). Notably, some variants are found across different clades and in mixed or mosaic clades (Fig. 3b,c), indicating that centromere variants can spread across populations through sex. This is further supported by the observation that strains can carry a combination of two different variants and that some centromere variants are present heterozygously (Fig. 3d). By including sexual cycles in our simulations, we show that full centromere transitions can spread through populations more easily (Fig. 3e). Next, we investigated the underlying mutational mechanisms driving the emergence of these new centromere variants. Remarkably, most variant centromeres seem to have expanded through a microhomology-mediated mutational mechanism. Aligning longer variants with their most similar short counterparts reveals that the newly inserted sequences are exact copies of short stretches of the original sequence (Fig. 3f). This is not only true for centromere variants in *S. cerevisiae* but also for variants detected in other Saccharomycetaceae (Extended Data Fig. 5b). One hypothesis is that these small homologous insertions are the result of stalled replication forks[23], which is a frequent occurrence at budding yeast centromeres[24]. Alternatively, they could have arisen through repair of double-strand breaks near centromeres during meiosis[25].

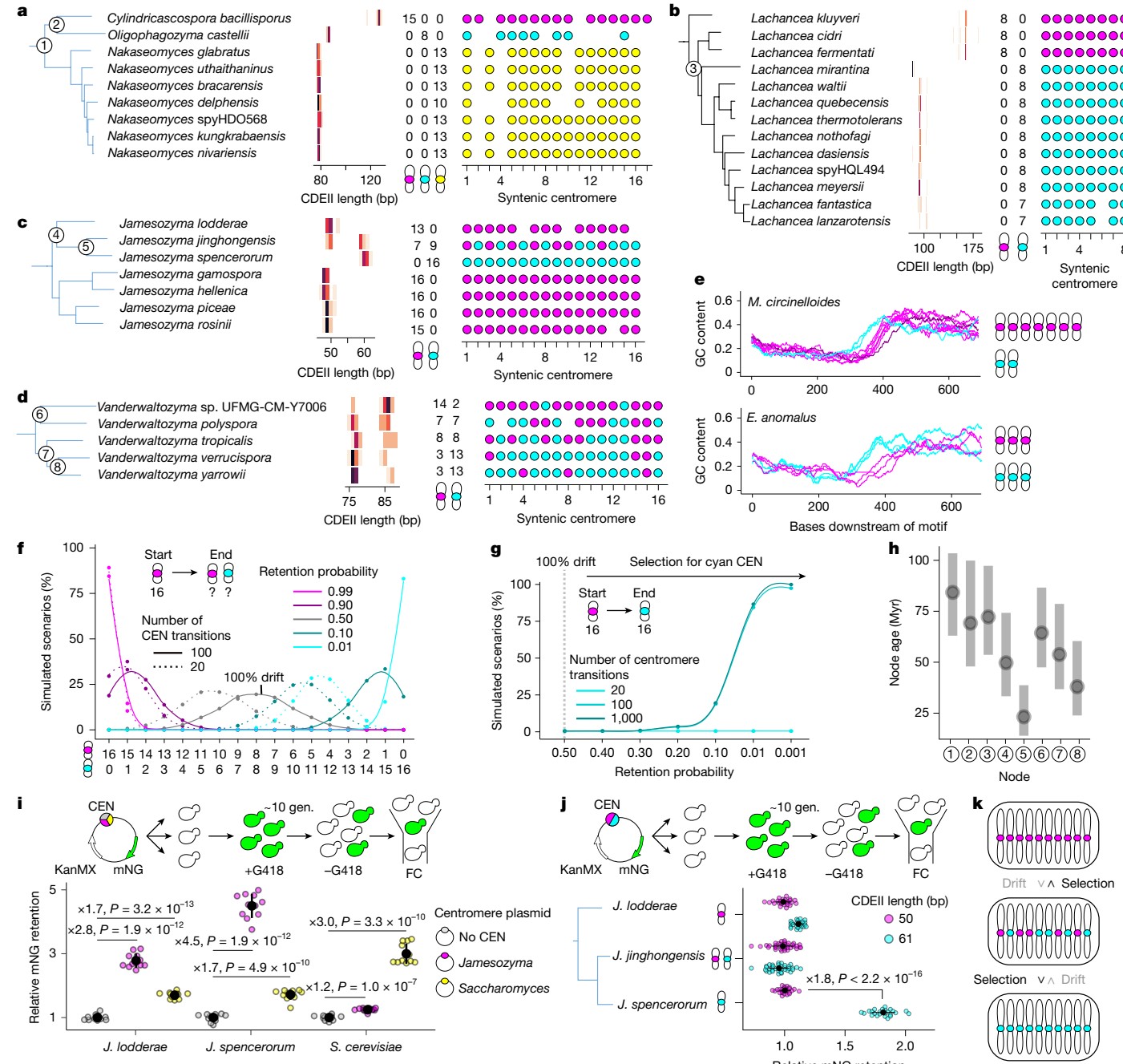

**Fig. 2 | Centromeres transition gradually through drift and selection.**
**a**–**d**, CDEII length across species in the *Cylindricascospora*, *Oligophagozyma*
and *Nakaseomyces* genera (**a**), the *Lachancea* genus (**b**), the *Jamesozyma* genus
(**c**) and the *Vanderwaltozyma* genus (**d**). Magenta, cyan and yellow indicate
different centromere types, as defined by different CDEII lengths. The total
number of each centromere type and their syntenic locations are indicated on
the right. The numbered nodes represent full or partial centromere transition
points. **e**, GC content of AT-rich centromere regions in two Mucoromycota
species, with colours indicating potential within-genome heterogeneity in the
length of this region. *E. anomalus*, *Ellisomyces anomalus*. **f**,**g**, Simulations of
centromere (CEN) transitions. **f**, Starting with 16 magenta centromeres, the
plot shows the proportion of 10,000 simulations for all possible final scenarios
(*x* axis, number of magenta and cyan centromeres). **g**, Specifically focusing
on the full transition scenario (16 magenta to 16 cyan), the *y* axis shows the

proportion of 10,000 simulations resulting in a full transition across 7 different
retention probabilities (*x* axis). **h**, Estimated node age of the nodes indicated
in **a**–**d**. Bars represent 95% confidence intervals. **i**,**j**, Plasmid retention assays.
Experiments were repeated once with similar results. **i**, Within-species relative
retention rate of a plasmid carrying no centromere (grey), a *Jamesozyma*
centromere (magenta) or a *Saccharomyces* centromere (yellow). Mean and s.d.
(black circles and lines) are based on $n = 12$ biologically independent samples
and were compared using a two-tailed Student's *t*-test. KanMX confers
G418 resistance; mNG, mNeonGreen; FC, flow cytometry; gen., generation.
**j**, Relative retention rate of a plasmid with a 50-bp CDEII (magenta) versus one
with a 61-bp CDEII (cyan) in three *Jamesozyma* species. Mean and s.d. (black
circles and lines) are based on $n = 24$ biologically independent samples and
were compared using a two-tailed Student's *t*-test. **k**, Schematic of the
proposed evolutionary model for centromere transitions.

Outside of the budding yeast clade, we observe similar patterns. *Mucor
circinelloides* genomes typically contain one set of ten distinct core
centromeres, although a subset of isolates are heterozygous diploids

with two complete sets (Fig. 3g). Together, these observations exem-
plify how centromere variants can spread and combine through sex
(Fig. 3h).

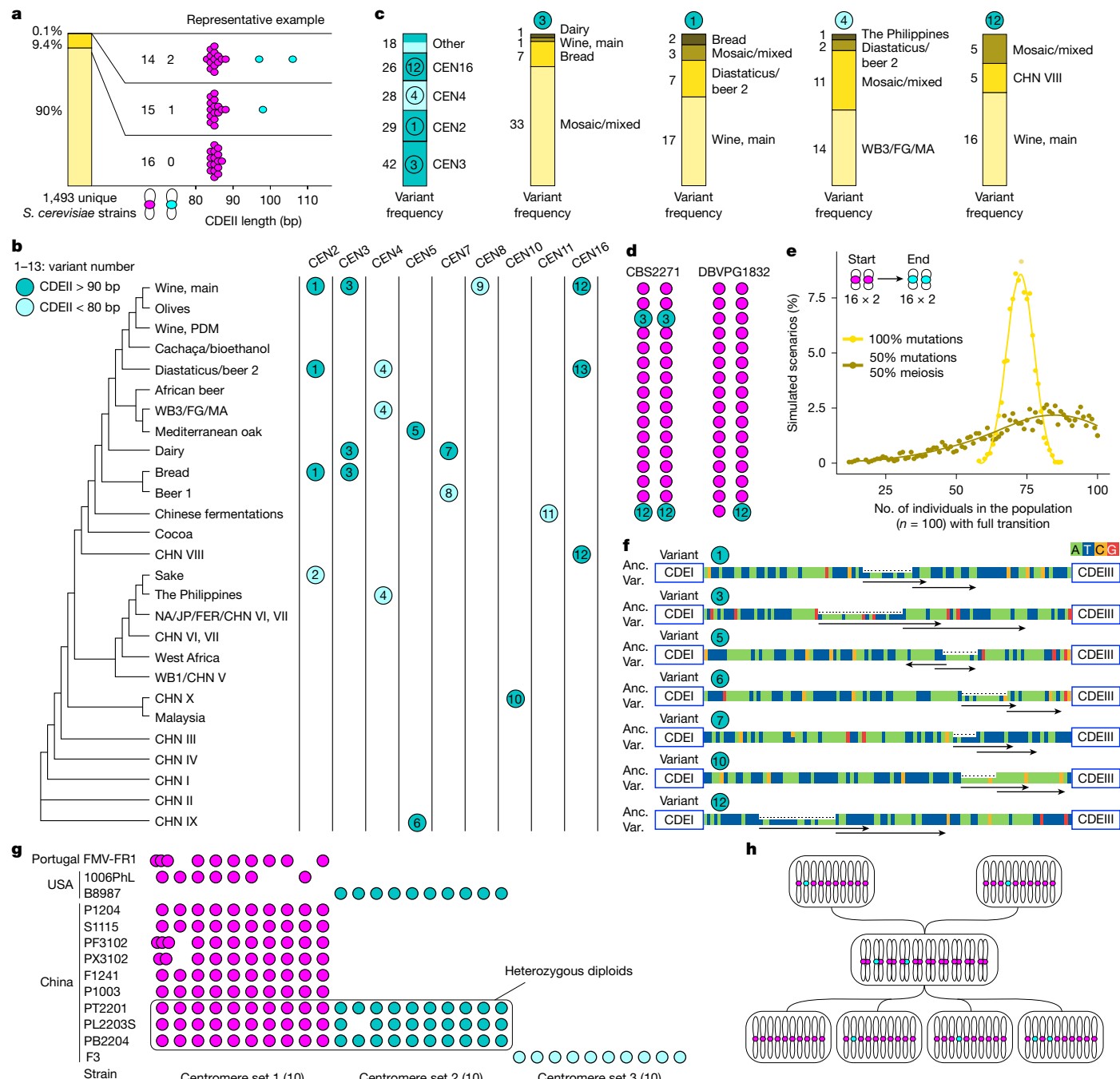

**Fig. 3 | Centromere variants spread through populations through sex.**
**a**–**c**, Centromere CDEII variants in *S. cerevisiae* populations. **a**, Proportion of *S. cerevisiae* strains with 'regular' centromeres (magenta, 80–90-bp CDEII) and variant centromeres (cyan). **b**, Distribution of variant centromeres throughout the natural *S. cerevisiae* population, mapped onto a phylogenetic tree. Light blue circles denote variants with CDEII < 80 bp; teal circles denote those with CDEII > 90 bp. Numbers in circles represent independent variants. Centromere IDs (for example, CEN2) correspond to chromosome IDs. Clade and tree structure data are sourced from ref. 47. **c**, Prevalence of the most frequent variants across different clades. **d**, Example of an *S. cerevisiae* strain (wine, main) with two variant centromeres (left) and a strain in which the variant centromere is heterozygous (right). **e**, Simulated full centromere transitions in diploid populations with and without meiosis. The simulation starts with 100 individuals, each with 16 × 2 magenta centromeres. At each step, a random

centromere in a random individual transitions (magenta ↔ cyan) with the retention probability of the magenta variant set to 1%. In the condition with meiosis, this transition is followed by a random reassortment of centromeres on the basis of population prevalence. The simulation was repeated 2,000 times for 500 steps. The *x* axis shows the number of individuals with full transitions, and the *y* axis shows the proportion of simulations resulting in a full transition. **f**, Mutations underlying CDEII length variation in *S. cerevisiae*. Variant sequences (Var.) are aligned with the most similar ancestral sequences (Anc.). Variant numbers match those in previous panels. Black arrows indicate identical sequences. **g**, Centromeres across different *M. circinelloides* isolates. Isolates contain one (haploids and homozygous diploids) or two (heterozygous diploids) sets of ten unique centromeres. **h**, Schematic illustrating how new centromere variants can spread and combine through sexual reproduction.

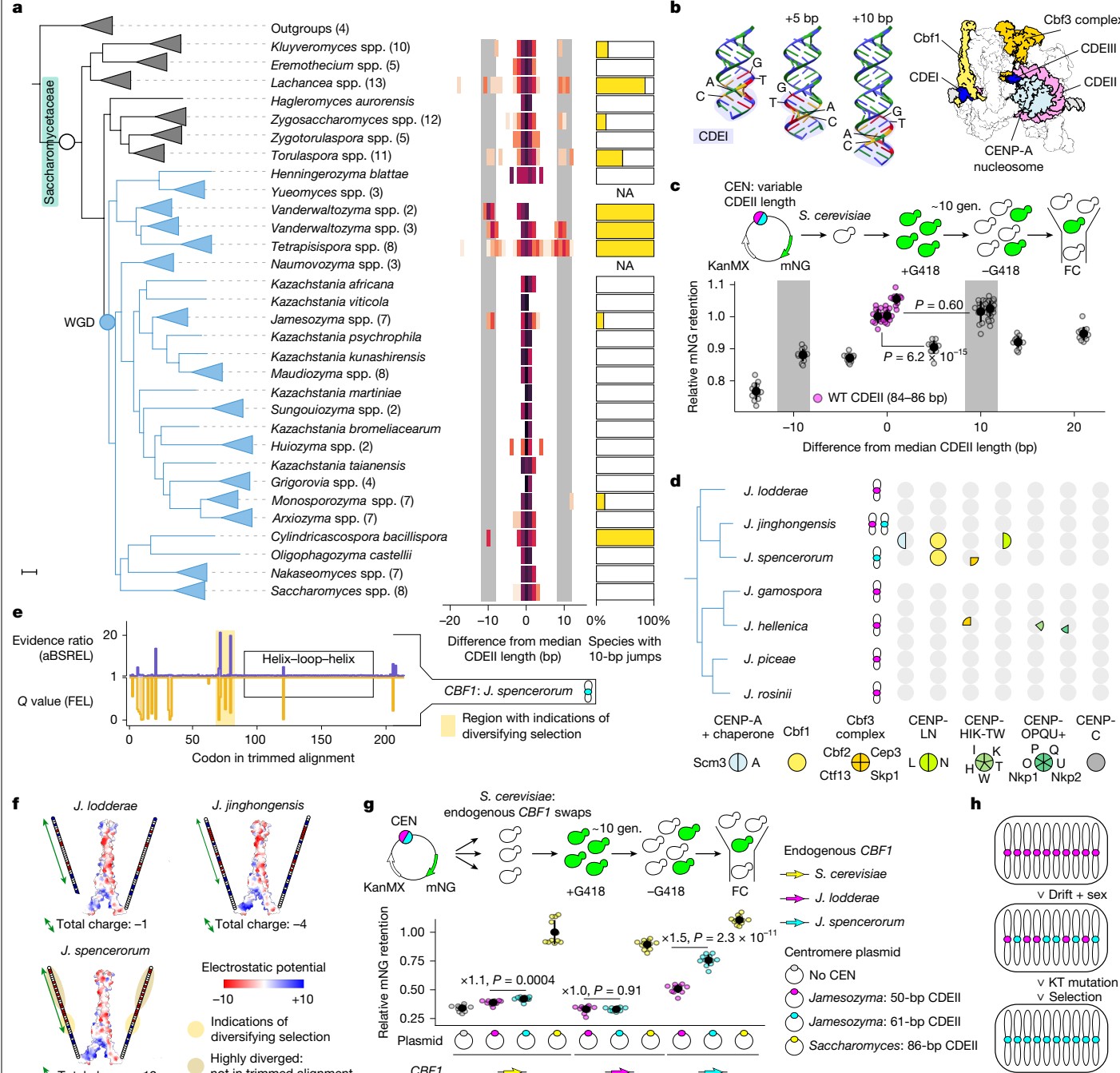

**Fig. 4 | Centromere transitions are constrained by the kinetochore interface. a**, CDEII length variation across Saccharomycetaceae. The difference from the median CDEII length and the proportion of species with 10 bp of CDEII length variation are shown for each clade. Genera are represented by triangles with the number of species in parentheses. NA, not applicable. Tree scale, 0.1. **b**, Relative orientation of the CDEI motif on pieces of DNA of different length and the structure of the inner kinetochore in *S. cerevisiae* highlighting DNA motifs (dark blue) and the AT-rich region (pink) together with the kinetochore proteins. Structure based on ref. 27. **c**, Plasmid retention assays with different CDEII length in *S. cerevisiae* (repeated once with similar results). Mean and s.d. (black circles and lines) are based on *n* = 12 biologically independent samples and were compared using a two-tailed Student's *t*-test. WT, wild type. **d**, Branches in the *Jamesozyma* genus with evidence of episodic diversifying selection for different inner kinetochore proteins. Adaptive branch-site random effects likelihood

(aBSREL) *P* values from left to right, top to bottom: 5.7 × 10⁻⁴, 0.031, 0.014, 1.4 × 10⁻⁶, 3.6 × 10⁻³, 8.8 × 10⁻³, 0.042 and 3.4 × 10⁻⁴. **e**, Evidence ratio (aBSREL) and *Q* value (contrast fixed effects likelihood (contrast-FEL)) across the trimmed alignment of *CBF1* in *J. spencerorum*. **f**, Predicted structures and electrostatic potential of Cbf1 dimers from *J. lodderae*, *J. jinghongensis* and *J. spencerorum*, predicted using AlphaFold2. Only segments with a predicted local distance difference test (pLDDT) > 70 are displayed as volumes. Blue circles represent positively charged residues, and red circles represent negatively charged residues. **g**, Plasmid retention assay in *S. cerevisiae* (performed once), with different combinations of endogenous *CBF1* replacements and centromere plasmids. Means were compared using a two-tailed Student's *t*-test. *n* = 11 biologically independent samples. Mean and s.d. are indicated by black circles and lines. **h**, Schematic of the proposed evolutionary model for centromere transitions. KT, kinetochore.

## Centromere–kinetochore coevolution

We next asked whether centromeres can transition to any new state or whether there are constraints on the types of changes that can occur. Although the centromere atlas in Fig. 1 shows that point centromeres and CDEII length in particular can vary extensively over longer evolutionary timescales, the examples shown in Figs. 2 and 3 suggest that transitions can only occur through a select set of mutations. In both examples, most mutations lead to an approximately 10-bp jump in CDEII length. The ability to tolerate such jumps in centromere length seems to be present across the point centromere clade, even though the expression of this trait varies between species (Fig. 4a and Extended Data Fig. 6). Although some genera, such as *Lachancea* and *Vanderwaltozyma*, show a consistent mix of CDEII lengths, others such as *Saccharomyces* only show the approximately 10-bp jumps in a proportion of the population (Fig. 3). As nucleosomal DNA twists at about 10.2 bp per turn[26], we hypothesize that an approximately 10-bp jump in CDEII length ensures that the CDEI motif remains in the orientation necessary for Cbf1 to properly bind the motif and interact with other components of the kinetochore complex (Fig. 4b). Indeed, by measuring the segregation efficiency of centromeres with different CDEII lengths in *S. cerevisiae*, we found that variants that are approximately 10 bp longer than wild-type centromeres show no segregation defects whereas centromere variants that are only 5 bp longer are lost more readily (Fig. 4c and Extended Data Fig. 7a). Similar to what we observed in natural isolates (Fig. 3), *S. cerevisiae* seems to tolerate longer centromere variants better than shorter variants. Together, these observations are consistent with a model that the kinetochore interface dictates which centromere variants can be tolerated and subsequently spread through populations.

Finally, although this accounts for how new neutral centromere variants can spread through drift, this does not explain how new variants might become more efficient and favoured by selection, as observed in *Jamesozyma* spp. in Fig. 2j. To explore whether this might be the result of adaptation of the kinetochore machinery itself, we tested the inner kinetochore proteins in the genus for evidence of positive selection. Coincident with centromere transitions, Cbf1, the protein that binds the CDEI motif (Fig. 4b), shows evidence of episodic diversifying selection (Fig. 4d). Most of the signal comes from a small segment of the N-terminal tail of the protein, close to the bHLH leucine zipper domain that binds CDEI[27] (Fig. 4e). The N-terminal tail of Cbf1 is a fast-evolving disordered region, the removal of which only leads to a minor reduction in segregation efficiency in *S. cerevisiae*[28]. In *J. spencerorum*, the species that recently underwent a complete centromere transition and in which the new centromere variant became the more efficient variant, the mutations resulted in a new acidic patch close to the zipper domain (Fig. 4f and Extended Data Fig. 7b), potentially influencing the overall orientation of Cbf1 through interactions with other proteins of the kinetochore. To test whether these different Cbf1 variants can indeed alter which centromere variants are preferentially retained, we replaced the endogenous *CBF1* gene in *S. cerevisiae* with different *Jamesozyma* variants and measured the relative retention rate of plasmids with either short or long *Jamesozyma* centromeres. Remarkably, the *J. spencerorum* Cbf1 protein made long *Jamesozyma* centromeres significantly more efficient than short centromeres (Fig. 4g), suggesting that mutations in this protein might indeed explain why the long variant acquired a selective advantage and reached fixation in the genome. Our findings suggest that coevolution between the kinetochore and the evolving centromere sequences could be an important driver behind centromere transitions (Fig. 4h).

## Discussion

With our new tool for point centromere annotation, we expanded the limited repertoire of available tools for clade-specific in silico centromere prediction[29–32] and generated a clade-wide overview of point centromere diversity. We showed that centromeres transition progressively through a combination of drift, selection and sex to new states that are compatible with the kinetochore interface. We propose a model in which the initial fate of a new centromere variant is largely determined by its compatibility with the existing kinetochore machinery. If the variant can establish a sufficiently stable interaction with the kinetochore, it can spread in the population through neutral processes such as drift and sexual reproduction. Subsequent changes in the segregation efficiency of different variants, either through modifications to the kinetochore machinery (for example, mutations in kinetochore proteins such as Cbf1) or through environmental changes (for example, temperature changes impacting microtubule dynamics) could then create selective pressures that ultimately drive full centromere transitions. Indeed, several kinetochore proteins show signatures of positive selection, some of which correlate with evolving centromeric features[1,33–37].

Smaller centromeres are found across the eukaryotic tree, especially in clades that have thus far received much less attention from the centromere community (Fig. 1b), nor are point centromeres the only type of centromeres in which DNA motifs are found. Many complex centromeres still contain motif-like elements that are bound by specific kinetochore proteins. Many mammalian centromeres contain CENP-B boxes[38,39], several fungi with regional centromeres still contain motifs[40], and motifs have also been found in diatom centromeres[41]. Together with our parallel observations in the Mucoromycota, we expect that, as the quality and density of genome assemblies continues to explode, our predictions could soon be experimentally validated in clades with complex centromeres across eukaryotes.

One popular model used to explain rapid centromere evolution is the centromere drive model, which proposes that the most efficient centromere variants are preferentially passed on to the next generation during asymmetric female meiosis, despite their potential fitness costs[42]. However, this model may not fully explain centromere evolution in the Saccharomycetaceae, as many species in this clade strictly undergo male meiosis in which (in the absence of other meiotic drivers[43]) all four meiotic products are viable. Moreover, our results indicate that differences in segregation efficiency during mitosis can also vary significantly between species. In unicellular organisms, both sexual (or germline) and asexual (or somatic) variation is passed on to the next generation. Our work underscores that centromere evolution is in fact the result of an interplay of various factors, including drift and selection during both mitosis and meiosis, along evolutionary trajectories that are constrained by the kinetochore interface.

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

## Methods

### Genomes

Genome assemblies for each of the 138 Saccharomycetaceae species, 4 outgroup species (*W. anomalus*, *C. albicans*, *P. kudriavzevii* and *Y. lipolytica*), Saccharomycodaceae and Mucoromycota were downloaded from the National Center for Biotechnology Information (NCBI) (Supplementary Data 5). Two additional Mucoromycota genomes were downloaded from the JGI (Supplementary Data 5). To explore *S. cerevisiae* intraspecies centromere variation, 2,737 assemblies were downloaded from the NCBI and from the supplemental data of 4 studies[48–51] (Supplementary Data 6). As some strain backgrounds (for example, S288c) are overrepresented in this dataset, we limited our selection to 1,493 unique strains with phylogenetic information for Fig. 3. Regardless, the overall frequency of variant centromeres was very similar between the reduced and full datasets (Extended Data Fig. 5a).

### Point centromere annotation pipeline

**PCAn pipeline.** FIMO[52] from the MEME suite (v.4.11.2)[53] was used to search each genome assembly with a chosen 26-bp-long CDEIII motif (see below), using a threshold ranging from $1.0 \times 10^{-3}$ to $1.0 \times 10^{-7}$. The coordinates of the hits were used to compile a list of 250-bp-long sequences that contain the CDEIII motif plus 224 bp upstream. Subsequently, this output was searched again using FIMO and a chosen 8-bp-long CDEI motif with a less-restrictive threshold of $1.0 \times 10^{-2}$. The coordinates of these hits were used to compile a list of sequences that contain a CDEI motif on one end and a CDEIII motif at the other end. For each of these sequences, the length and AT percentage of the intermediary CDEII motif were calculated. Every sequence was assigned a combined score on the basis of the two FIMO hit scores and the CDEII AT percentage. Sequences were sorted on the basis of the combined score, and only the top 50 sequences were retained. Next, we determined the median CDEII length of the top 5 sequences and removed sequences with lengths differing by more than 30 nucleotides from the median. Finally, after removing sequences with a low CDEII AT content (<70%) and removing duplicates (sometimes found on small contigs of lower-quality assemblies), we removed sequences in which both CDEII length and AT content differed too much from the median (≥median length ± 10 nucleotides and <median AT percentage - 7, respectively).

**Motif construction and selection.** CDEI and CDEIII sequences from ref. 9 were used to produce the initial search motifs, using the sites2meme command from the MEME suite (v.4.11.2)[53]. Newly discovered sequences were each verified using synteny and used to make new motifs and perform more sensitive searches in specific clades or species. The final PCAn pipeline uses the best-performing custom motifs and thresholds for each clade or species.

**Synteny checks.** Synteny checks were performed by identifying proteins encoded 10 kb upstream and downstream of centromere hits. After using contig and coordinate information of each centromere hit to extract the approximately 20-kb region around each potential centromere, open reading frames (ORFs) were identified using the getORFProteins function from ORFFinder Python (v.1.8) (minimum_length = 525, remove_nested = True, return_loci = True)[54]. ORFs were then BLASTed against the *S. cerevisiae* proteome using a local version of NCBI blastp (v.2.13.0+, default parameters)[55]. Finally, *S. cerevisiae* protein identifiers were matched to the ancestral Saccharomycetaceae gene order identifiers from the Yeast Gene Order Browser[56], which were used for visualization.

**Adapting PCAn for Mucoromycota.** Core centromeres in Mucoromycota were predicted using a slightly altered version of PCAn. Only one motif search was conducted, starting from the 41-bp-long motif identified in ref. 12. The coordinates of the hits were used to compile a list of sequences containing the motif and an additional 750 bp downstream. As *Mucor* core centromere sequences are often found on very short contigs (probably owing to assembly issues because of the flanking retrotransposons), the sequences were often shorter than 750 bp. Next, we visualized the average GC content of the downstream sequence, using a moving window with a window size of 100 bp. This proved to be a straightforward yet reliable method to identify bona fide core centromeres. As we identified motifs in other Mucoromycota species, the search motifs were iteratively changed using the newly discovered sequences, and new searches were performed until no additional centromeres were found.

### Inner kinetochore protein searches

On the basis of a literature review and the list used in ref. 27, we selected 21 inner kinetochore proteins in *S. cerevisiae*. We used tblastn (BLAST suite v.2.5.0+, default parameters)[55] to retrieve the coordinates for the ORFs in the genomes of the 137 other Saccharomycetaceae. We rejected all hits with an E-value greater than $1 \times 10^{-10}$. For the remaining hits, we then used the getORFProteins option (minimum_length = 210, remove_nested = True, return_loci = True) in ORFFinder Python (v.1.8)[54] to identify all possible ORFs within 5 kb on either side of the starting coordinate. We selected the ORFs that contained the midpoint of the selected 10-kb scaffold. We further selected only those ORFs that, on performing blastp against the *S. cerevisiae* proteome, returned the correct starting protein as the best hit. Finally, for each instance for which this approach failed to retrieve homologues, we manually checked and confirmed the presence or absence of homologues using phylum-specific homology searches.

### Species tree construction

**Selecting marker proteins.** We inferred the species tree for the set of 142 species (138 Saccharomycetaceae and 4 outgroup species: *W. anomalus*, *C. albicans*, *P. kudriavzevii* and *Y. lipolytica*) using 1,403 marker genes identified and published in an earlier study[57]. From this set of 1,403 markers, 113 markers were removed, as they did not have a corresponding protein homologue in *S. cerevisiae*. We mapped the markers to the *S. cerevisiae* reference proteome using phmmer from the HMMER suite of tools[58] and removed 13 markers that returned the same best hit as other markers in the dataset, resulting in a set of 1,277 markers.

**Retrieving homologues for marker proteins.** We constructed BLAST databases using makeblastdb (BLAST suite v.2.5.0+)[55] for each of the 142 genomes. We used tblastn (default parameters) to retrieve the ORF coordinates for the 1,277 markers from the 142 genomes. We used the getORFProteins option in ORFFinder Python (v.1.8)[54] to identify the entire ORF for each potential homologue with an E-value < $1 \times 10^{-10}$ and only selected those homologues that returned the corresponding marker protein in the *S. cerevisiae* proteome as the best hit in a reverse blastp search. At this stage, we excluded 7 more markers for which we were only able to identify reciprocal best hits in less than 50% of the selected 142 species.

**Constructing gene trees, removing outliers and reconstructing and dating the species tree.** For each of the 1,270 sets of homologues, we used MAFFT (v.7.505)[59] with the E-INS-i option to align sequences, trimAl v.1.4.rev15 build[2013-12-17][60] with the -gappyout option to remove phylogenetically noisy positions and FastTree v.2.1.11 Double precision (No SSE3)[61] with options -spr 4 -mlacc 2 -slownni -n 1000 -gamma to build ML gene trees. We analysed these trees using ETE3 (ref. 62) in Python to identify and remove branches in the tree with branch lengths greater than 20 times the median branch length. In cases in which the median branch length was less than $1 \times 10^{-8}$, we manually inspected the trees and alignments to remove outliers. These steps were repeated until no outliers were found.

We concatenated the 1,270 alignments obtained after outlier removal into one supermatrix alignment with 672,500 positions using Goalign[63]. We used IQ-TREE multicore (v.2.2.0.3)[64] with options -B 1000 -alrt 1000 --boot-trees --wbtl -m LG + G4 -mwopt --threads-max 24 -T AUTO to build an ML tree using the LG model[65] with 4 rate categories (LG + G4) with 1,000 ultrafast bootstraps[66]. We used the LG + G4 model, as it was the selected model for 661 of 1,270 marker genes.

To estimate divergence times in a tractable manner, we randomly subsampled the supermatrix alignment, extracted three sets of 10,000 sites and inferred ML trees using IQ-TREE as described earlier. We inferred the time tree for these three trees using the RelTime-ML method[67] implemented in MEGA11 (ref. 68). We adopted two well-estimated ranges of divergence from two internodes as calibration points: the *S. cerevisiae*–*Saccharomyces uvarum* split (14.3–17.94 Ma) and the *S. cerevisiae*–*Kluyveromyces lactis* split (103–126 Ma)[16,46].

### Sanger sequencing of *J. jinghongensis* centromeres
To Sanger sequence each of the 16 *J. jinghongensis* centromeres, we designed 16 primer pairs to specifically amplify each centromere (oligonucleotide sequences can be found in Supplementary Data 7). We then isolated gDNA from *J. jinghongensis* (CBS 15232) using the MasterPure Yeast DNA Purification Kit from Lucigen and amplified each region by PCR. The PCR product was then sent for Sanger sequencing (Eurofins Genomics).

### Centromere transition simulations
**Haploids and diploids without meiosis.** Each experiment was initialized with one individual carrying 16 centromeres of type A. Next, the random number generator of numpy (seed, 19,680,801) was used to generate a random number from 0 to 15 to pick one random centromere. After this, a second random number between 0 and 999 was generated and compared with the retention probability to decide whether the chosen centromere would be forced to transition or not. The retention probability always refers to how probable it is to retain the original type A. For example, if the chosen centromere is of type A and the retention probability is 0.99, numbers from 0 to 989 will lead to retention of type A and numbers from 990 to 999 will lead to a transition to type B. Similarly, if the chosen centromere is of type B, numbers from 0 to 989 will lead to transition to type A and numbers from 990 to 999 will lead to retention of type B. This process was repeated 20, 100 or 1,000 times for different retention probabilities. The simulations were repeated 10,000 times. Simulations for diploids were carried out in the exact same manner but were initialized with one individual carrying 32 centromeres of type A.

**Diploids with and without meiosis.** Each experiment was initialized with 100 individuals carrying 2 × 16 centromeres of type A. Each iteration was composed of a mutation step, similar to what is described above for haploids, followed by a meiosis step for the condition with meiosis. For the mutation step, numpy (seed, 19,680,801) was used to pick a random individual in the population and then a random chromosome pair and finally a random chromatid. Next, exactly the same as for haploids, another random number was generated and compared with the retention probability to decide whether the chosen centromere would be forced to transition or not. For the condition with meiosis, each individual in the population was then allowed to undergo meiosis: for each chromosome, a random chromatid was retained and the other chromatid was randomly picked from the population. This process was repeated 500 times, with and without meiosis. The simulations were repeated 2,000 times.

### Plasmid retention assays
To measure the relative retention rates of plasmids containing different centromere variants, we used the same principle as that used in ref. 69. Apart from components necessary for propagation in *Escherichia coli*, each plasmid contained an autonomously replicating sequence shown

to function across different Saccharomycetaceae[70], a KANMX6 cassette for selection in yeast, a pADH1-mNeonGreen-tADH1 construct for constitutive expression of mNeonGreen and a variable centromere sequence. Plasmids were made using NEBuilder HiFi Assembly (NEB) or Q5 site-directed mutagenesis (NEB) and verified by whole-plasmid sequencing (Plasmidsaurus). Oligonucleotides and cloning strategies for each plasmid can be found in Supplementary Data 7 and 8. To swap the endogenous *CBF1* gene with *Jamesozyma* variants in *S. cerevisiae*, we constructed *Jamesozyma* promoter-*CBF1*-terminator-hphMX6 cassettes and introduced the cassettes into the endogenous *S. cerevisiae* *CBF1* locus. Plasmid sequences can be found on the Figshare repository accompanying this paper[71]. Plasmids and cassettes were transformed into *J. lodderae* (CBS 2757), *J. jinghongensis* (CBS 15232), *J. spencerorum* (DBVPG 6746) and *S. cerevisiae* (BY4741) using the standard LiAc-based transformation protocol for budding yeast[72]. All strain information can be found in Supplementary Data 9.

For the retention assays, cells were grown overnight in 10 ml YPAD (1% (wt/vol) yeast extract (BD), 2% (wt/vol) peptone (BD), 2% (wt/vol) glucose (Merck), 40 mg l[−1] adenine sulphate (Sigma)) supplemented with G418 disulfate (Carl Roth): 800 µg ml[−1] for *Jamesozyma* spp. and 400 µg ml[−1] for *S. cerevisiae*. Except for *J. jinghongensis*, which was grown at 25 °C, all other species were grown at 30 °C throughout the experiment. The next morning, the $OD_{600}$ was measured, cultures were washed with YPAD and diluted to $OD_{600}$ = 1 in YPAD. A sample was taken to determine the proportion of fluorescent cells at $t_0$. The cultures were then diluted 1:256 in YPAD (200 µl in 50 ml), and 150 µl of these cultures was pipetted into wells of a 96-well plate. Cells were grown for 24 h (about ten generations), after which the proportion of fluorescent cells was measured by flow cytometry on an Accuri C6 (BD Biosciences) or a FACSymphony A3 (BD Biosciences) system (minimum of 30,000 singlets). Forward and side scatter were used to select singlets, and the fluorescence distribution of the $t_0$ sample was used to set the gate distinguishing fluorescent from nonfluorescent cells. Using those proportions, the relative mNeonGreen retention was determined by performing within-species normalization by dividing each data point by the mean of the lowest-performing plasmid in the same species and experiment (Fig. 2) or by dividing by the mean of the wild type (Fig. 4).

To calculate false positive ratios of plasmid retention assays, we prepared our samples in exactly the same manner as described above. We then sorted fluorescent single cells into PBS using a FACSDiscover S8 Cell Sorter (BD Biosciences) and plated approximately 267 cells per plate on either YPD or YPD with G418 (200 µg ml[−1]) (six plates per condition per genotype). The plates were incubated for 2 days at 30 °C, except for those with *J. jinghongensis*, which were incubated for 2 days at 25 °C. Colonies were then manually counted to determine the false positive rate.

### Protein evolutionary analyses
For every *Jamesozyma* spp. and each of the inner kinetochore proteins identified above, we extracted the corresponding gene sequence using the same combination of tblastn (BLAST suite v.2.5.0+, default parameters)[55] and ORFFinder Python (v.1.8)[54]. Protein sequences were aligned using MAFFT (v.7.505)[59] (default parameters), and protein alignments were then converted to codon alignments using PAL2NAL (v.14)[73] (default parameters). Gaps and ambiguously aligned sites were removed using Gblocks v.0.91b[74], using -t = c; -b1 = $b; -b2 = $b; -b3 = 1; -b4 = 6; -b5 = h, with $b the number of sequences divided by two plus one, as applied in refs. 75,76. aBSREL (v.2.5)[77] and contrast-FEL (v.0.5)[78] from the HyPhy suite (using default parameters) were then used to search for signatures for episodic diversifying selection and differences in selective pressure at individual sites in *J. spencerorum*, respectively.

### AlphaFold2-Multimer modelling of Cbf1 dimers
We predicted structures for Cbf1 dimers from *J. lodderae*, *J. jinghongensis* and *J. spencerorum* using a local installation of ColabFold

1.5.5 (ref. 79) (--num-recycle 12 --num-ensemble 1 --model-type auto --save-pair-representations). We used the entire Cbf1 sequence as retrieved from the homology searches. ColabFold used MMseqs2 (ref. 80) on specific, clustered databases[81,82] to retrieve homologues for Cbf1 and construct alignments. AlphaFold2-Multimer[83] with 12 recycles was used to predict the structure of the Cbf1 dimer. The resulting structure predictions were visualized using UCSF ChimeraX (v.1.8)[84].

## Reporting summary

Further information on research design is available in the Nature Portfolio Reporting Summary linked to this article.

## Data availability

All data accompanying this paper can be found on Figshare (https://doi.org/10.6084/m9.figshare.c.7630151)[71]. Genomes were downloaded from the NCBI (https://www.ncbi.nlm.nih.gov/datasets/genome/), the JGI (https://jgi.doe.gov) and the supplemental data of four studies (refs. 48–51: https://doi.org/10.1038/s41586-018-0030-5, https://doi.org/10.1111/mec.13341, https://doi.org/10.1016/j.gene.2024.148722 and https://doi.org/10.1007/s00253-024-13267-3). Tables with all genome accession numbers (165 different species and 2,737 *S. cerevisiae* isolates) can be found in Supplementary Data 5 and 6 and on Figshare (https://doi.org/10.6084/m9.figshare.c.7630151)[71]. The TimeTree database (https://timetree.org) was used to retrieve the divergence time between *S. cerevisiae* and *K. lactis*[46]. The Yeast Gene Order Browser[56] (http://ygob.ucd.ie) was used for synteny checks. All materials are available upon request. Source data are provided with this paper.

## Code availability

PCAn and all other code used for this project are under an MIT license. The version of the code used for this study is v.1.0 and it can be found on Zenodo (https://doi.org/10.5281/zenodo.17293586)[19]. The most current version can found at https://github.com/JHelsen/point-centromere-detection.

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

**Acknowledgements** We thank R. Carvalho for helping us run samples for flow cytometry; T. A. Williams from the University of Bristol for his invaluable discussions and suggestions with regards to the species tree reconstruction; and M. Hays, B. Baum and P. Gönczy for invaluable critical feedback on the manuscript; we are also grateful for the resources and assistance of B. Ramasz, D. Gimenes and A. Cabrera Nuñez from the EMBL Flow Cytometry Core facility, EMBL IT services and the HPC cluster (https://doi.org/10.5281/zenodo.12785829). Funding was provided by the Life Science Alliance (Bridging Excellence Fellowship) (J.H.), NIGMS R35 GM131824 (G.S.), the European Molecular Biology Laboratory (J.H., K.R. and G.D.), the European Union (ERC, KaryodynEvo, 101078291) (J.H., K.R. and G.D) and the Joachim Herz Stiftung (Add-on Fellowship for Interdisciplinary Life Science) (K.R.). G.D. acknowledges the support of the EMBO Young Investigator Programme.

**Author contributions** Conceptualization: J.H., G.S. and G.D. Methodology: J.H., K.R., G.S. and G.D. Investigation: J.H., K.R., G.S. and G.D. Formal analysis: J.H. and K.R. Visualization: J.H. Funding acquisition: J.H., K.R., G.S. and G.D. Project administration: J.H., G.S. and G.D. Supervision: G.S. and G.D. Writing (original draft): J.H., K.R., G.S. and G.D. Writing (review and editing): J.H., K.R., G.S. and G.D.

**Competing interests** Open access funding provided by European Molecular Biology Laboratory (EMBL).

**Funding** Open access funding provided by Universität Potsdam.

**Additional information**
**Correspondence and requests for materials** should be addressed to Jana Helsen, Gavin Sherlock or Gautam Dey.

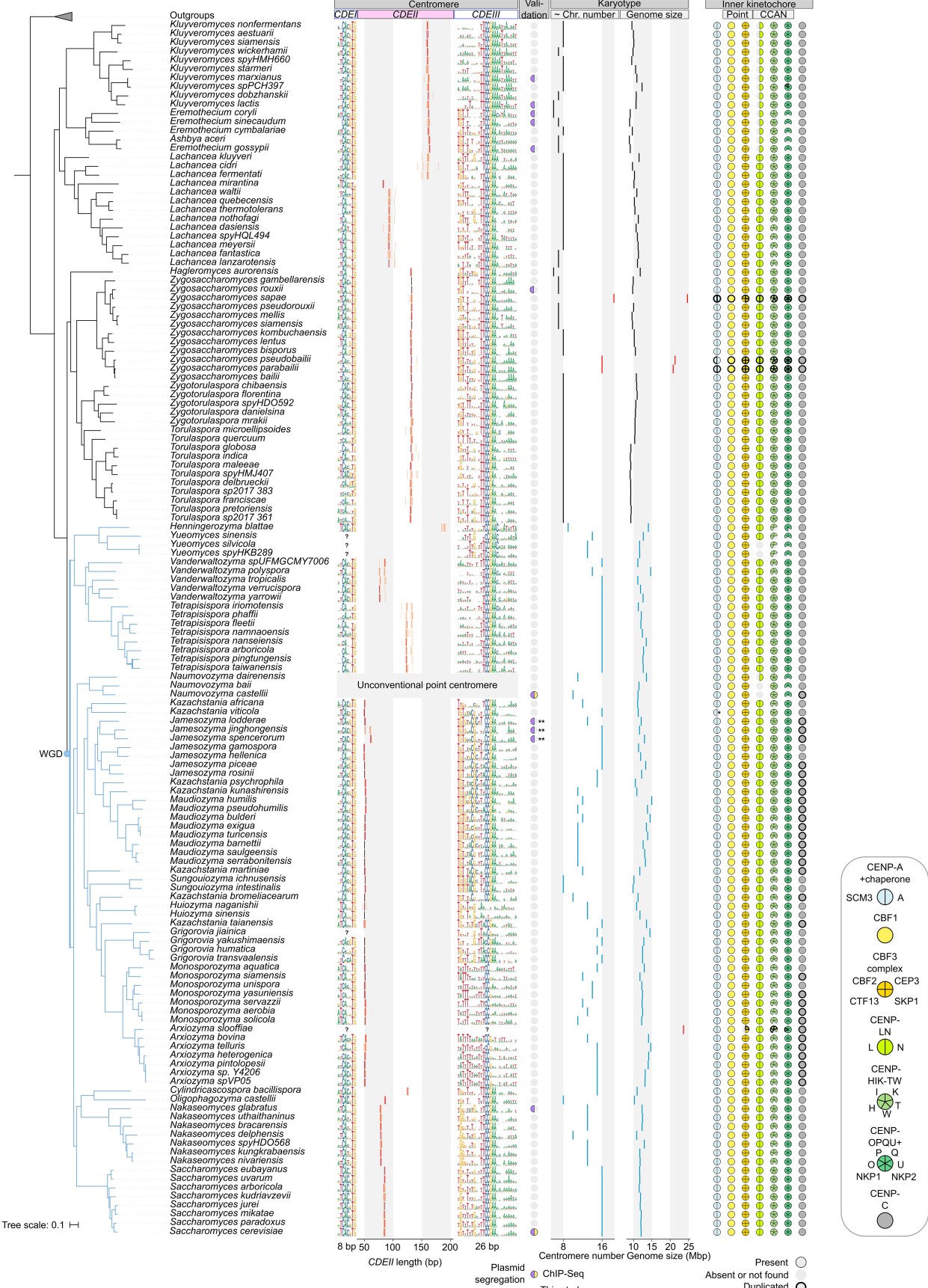

**Extended Data Fig. 1** | See next page for caption.

**Extended Data Fig. 1 | Uncollapsed tree with centromere sequences and inner kinetochore absence/presence protein profiles for 138 Saccharomycetaceae.** Species phylogeny was determined using a concatenation-based maximum likelihood analysis of 1,270 orthologous groups of proteins under a single LG + G4 model. Branches of species that emerged after the whole genome duplication (WGD) are colored in blue. Outgroup species: *Wickerhamomyces anomalus, Candida albicans, Pichia kudriavzevii*, and *Yarrowia lipolytica*. Centromere sequences are represented by DNA logos for CDEI and CDEIII, with graphs indicating CDEII length in between. The Validation column indicates which centromeres have been experimentally validated, either through plasmid segregation assays (purple) or ChIP-seq (yellow). Centromere numbers and genome size for each species are indicated in the middle panels with estimates for hybrid species indicated in red. CCAN: constitutive centromere-associated network. *CENP-A was not detected in *Kazachastania viticola*, but contigs in the genome assembly are broken up in the region where the gene is supposed to be. Hence, we believe its absence is an artifact of the assembly.

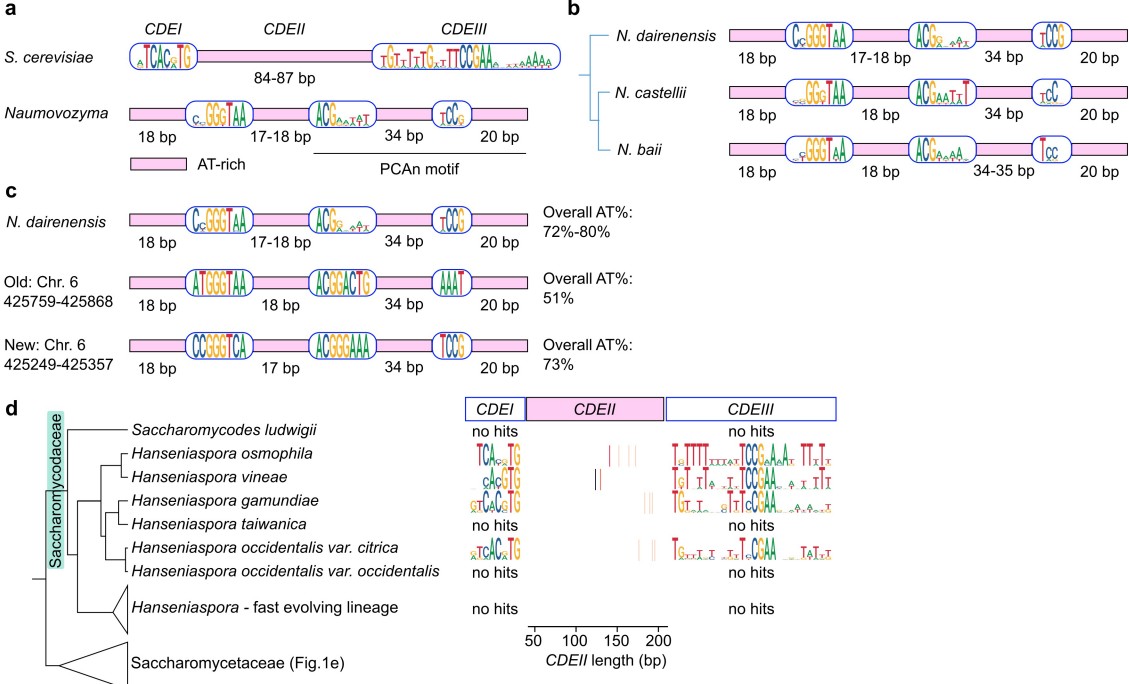

**Extended Data Fig. 2 | Unconventional point centromeres and proto-point centromeres.** (**a**) *Naumovozyma* spp. lack conventional point centromeres. Instead, they have point centromeres with unique, non-syntenic CDEs, indicating that they did not evolve from conventional point centromeres and have an independent evolutionary origin[45]. To see if our method can be adapted to also identify these types of point centromeres, we adapted PCAn and tried different versions of the motifs identified in ref. 45. In contrast to the conventional version of PCAn, what worked best was to use one long 66 bp motif instead of two. (**b**) Using this altered version of PCAn, we were able to correctly identify all ten centromeres in *N. castellii*. In *N. dairenensis*, we identified the correct number of centromeres (eleven), but on chromosome 6 we predicted a sequence different from ref. 45. (**c**). Since our hit is very close to the original prediction (and syntenic with the centromere found in *N. castellii*), and a much better match to the consensus motif, we propose that our hit is the 'real' centromere sequence of chromosome 6. For *N. baii*, PCAn only managed to detect 6 high-confidence syntenic hits. Either the motif in this species is too different to be picked up by our pipeline, or some centromeres relocated, or the quality of the assembly is insufficient. (**d**) Proto-point centromeres[10] picked up by PCAn. For four *Hanseniaspora* spp., PCAn picks up high confidence syntenic centromere sequences. The locations for *Hanseniaspora vineae* correspond to those recently identified by Hi-C in Haase et al.[10].

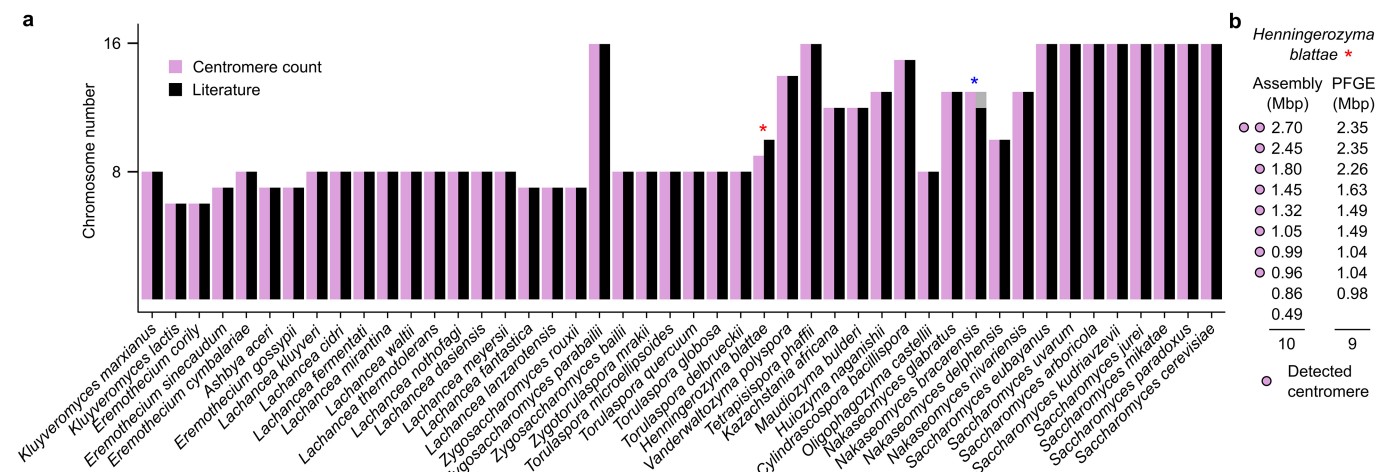

**Extended Data Fig. 3 | Predicted centromere numbers correspond to chromosome numbers reported in literature. (a)** Computed centromere counts (plum) versus chromosome numbers reported in literature (black). The blue asterisk indicates a case in which our pipeline identified 13 centromeres and the most recent assembly is unsure whether there are 12 or 13 chromosomes[85]. The red asterisk indicates one case in which the numbers are not identical.

This is further discussed in **(b)**, in which chromosome sizes from the chromosome-level assembly are compared to chromosome sizes observed in PFGE gels of the same strain (*Henningerozyma blattae* CBS 6284)[86]. Our chromosome number estimate corresponds with the estimate obtained using PFGE gels.

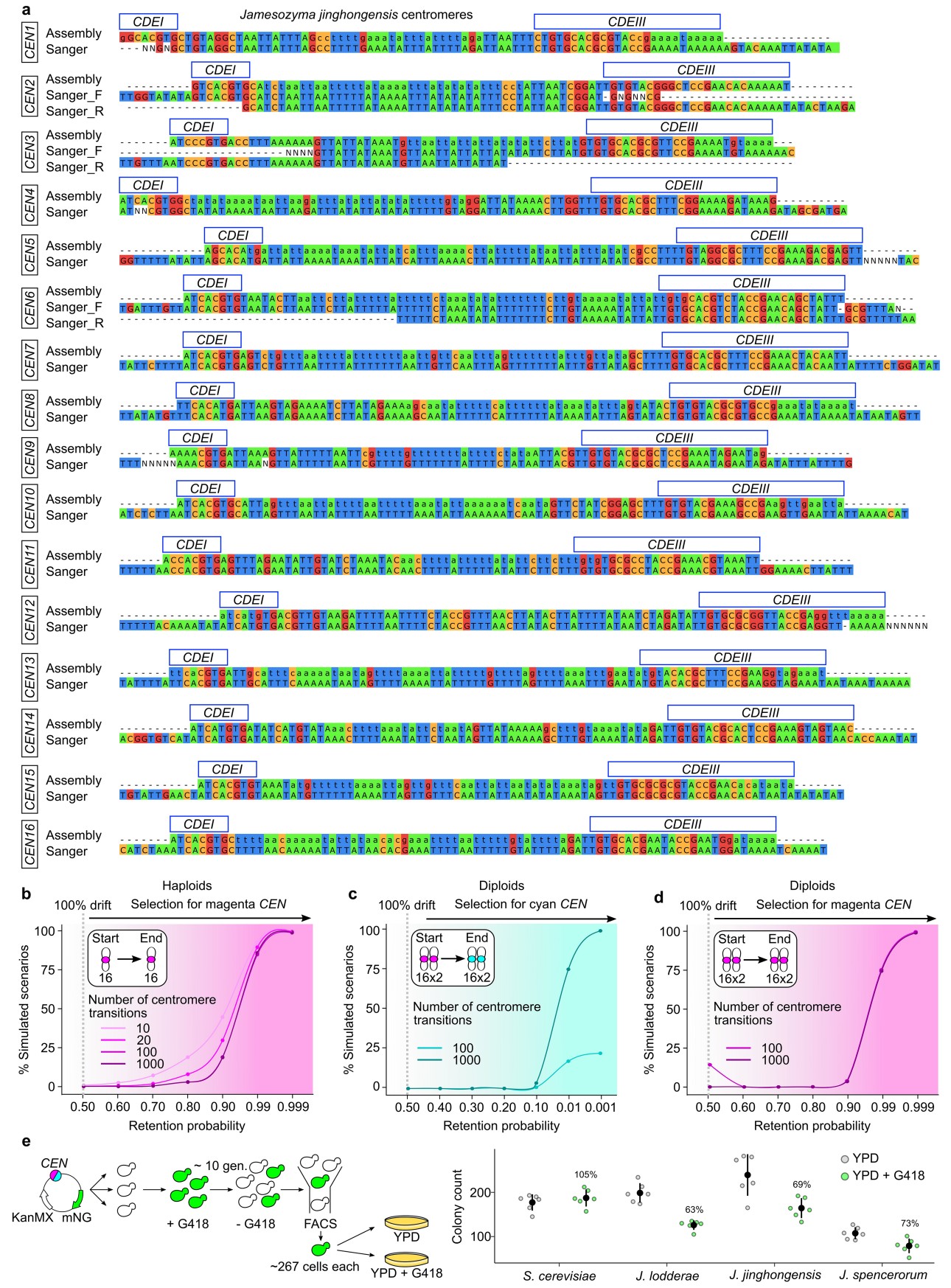

**Extended Data Fig. 4** | See next page for caption.

**Extended Data Fig. 4 | Centromere sequence validation, supplemental simulations, and false positive rate of the plasmid retention assay.**
(**a**) Aligned Sanger sequencing (Sanger) of predicted centromeres (Assembly) in *Jamesozyma jinghongensis*. (**b**) Simulated centromere transitions - full retention (haploids). Starting from 16 magenta centromeres, each transition, one random centromere was drawn and transitioned from magenta to cyan or cyan to magenta. The chance that this new variant is retained was then determined by the retention probability, where a 100% retention implies that magenta variants will always be retained (i.e., selection for magenta), a 50% retention rate implies that there is an equal chance the new variant is retained or lost (i.e., drift), and a 0% retention rate implies that magenta variants are never retained (i.e., selection for cyan). Here we specifically focus on the scenario of full retention (16 magenta centromeres remain 16 magenta centromeres). The *x* axis represents different retention probabilities and the *y* axis gives the proportion of the 10,000 simulations in which we observe a full transition for seven different retention probabilities. (**c**) Simulated centromere transitions - full transition (diploids). The simulations were done the same way as described for panel b but starting from 32 magenta centromeres and specifically focused on the scenario of full transitions (16×2 magenta centromeres to 16×2 cyan centromeres). (**d**) Simulated centromere transitions - full retention (diploids). The simulations were done the same way as described for panel c but specifically focused on the scenario of full retention (16×2 magenta centromeres to 16×2 magenta centromeres). (**e**) To determine the false positive rate of the plasmid retention assays, we grew the cells similarly as for any other plasmid retention experiment, but used FACS to sort fluorescent cells. Fluorescent cells were then plated on YPD and YPD + G418 (~ 267 cells each) and colonies were counted after two days. *n* = 6 biologically independent samples. Mean and s.d. are indicated by black circles and lines.

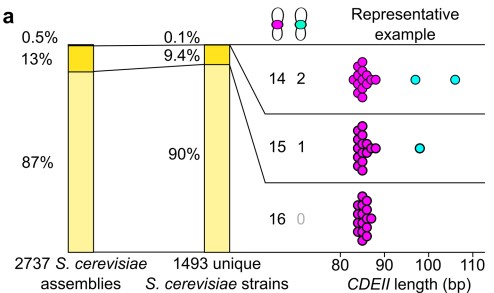

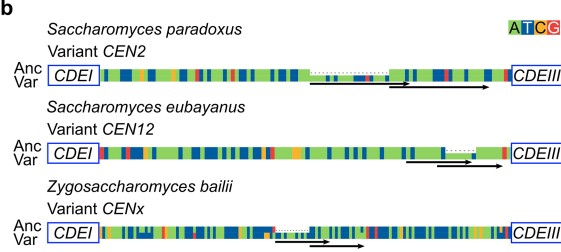

**Extended Data Fig. 5 | Overview of examined *S. cerevisiae* assemblies and microhomology in other *Saccharomyces* species.** (**a**) Proportion of *Saccharomyces cerevisiae* assemblies and unique strains with 'regular' centromeres (magenta, 80-90 bp CDEII) and variant centromeres (cyan). (**b**) Centromere variants with microhomology in species other than *S. cerevisiae*. Variant numbers correspond to the numbers indicated in the previous panels. Variant sequences (Var) were aligned with the most similar 'ancestral' sequences (Anc). Black arrows indicate identical sequences.

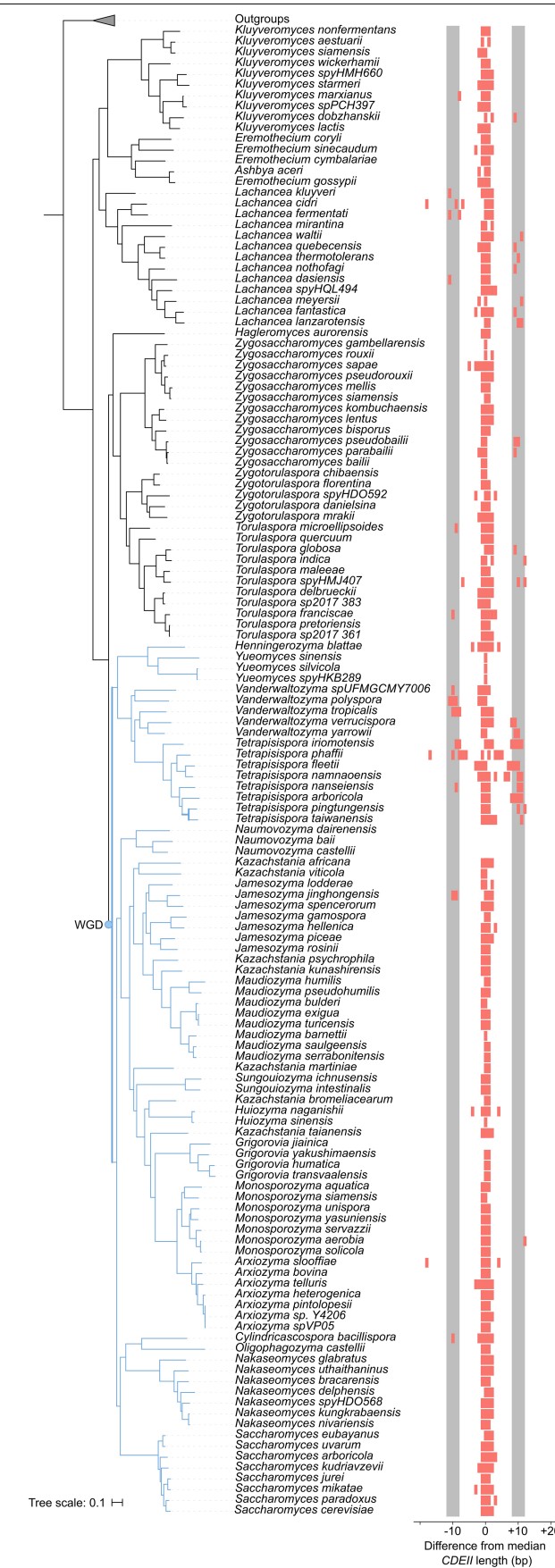

**Extended Data Fig. 6 | CDEII length variation profile across 138 Saccharomycetaceae.** Intragenomic CDEII length variation is represented by the difference from the median CDEII length. Branches of species that emerged after the whole genome duplication (WGD) are colored in blue.

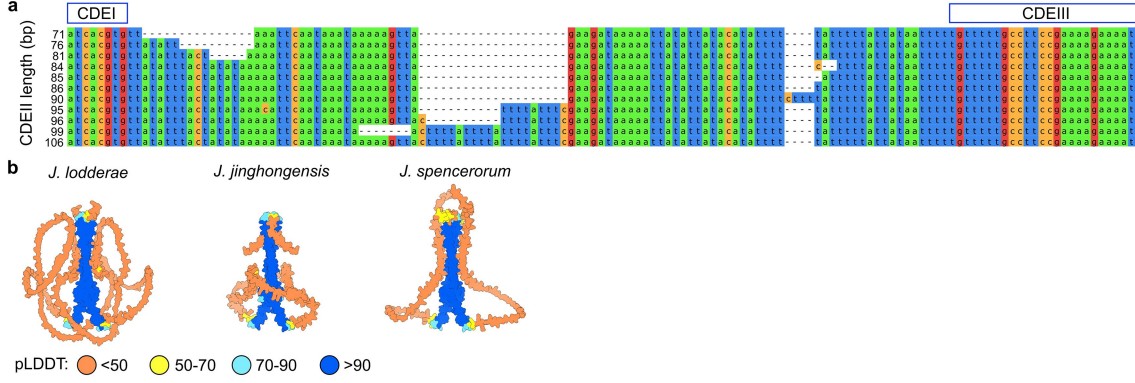

**Extended Data Fig. 7 | CDEII length variants and full structure predictions.**
(**a**) Alignment of centromere variants used in Fig. 4c. (**b**) Full Alphafold2
predictions of Cbf1 dimers. Structures of Cbf1 dimers for *J. lodderae,*
*J. jinghongensis* and *J. spencerorum*. Structures were predicted using Alphafold2
and colours represent different pLDDT values.

Gavin Sherlock
Gautam Dey

# Reporting Summary

## Statistics

For all statistical analyses, confirm that the following items are present in the figure legend, table legend, main text, or Methods section.

| n/a | Confirmed | |
|---|---|---|
| ☐ | ☒ | The exact sample size (*n*) for each experimental group/condition, given as a discrete number and unit of measurement |
| ☐ | ☒ | A statement on whether measurements were taken from distinct samples or whether the same sample was measured repeatedly |
| ☐ | ☒ | The statistical test(s) used AND whether they are one- or two-sided *Only common tests should be described solely by name; describe more complex techniques in the Methods section.* |
| ☒ | ☐ | A description of all covariates tested |
| ☒ | ☐ | A description of any assumptions or corrections, such as tests of normality and adjustment for multiple comparisons |
| ☐ | ☒ | A full description of the statistical parameters including central tendency (e.g. means) or other basic estimates (e.g. regression coefficient) AND variation (e.g. standard deviation) or associated estimates of uncertainty (e.g. confidence intervals) |
| ☐ | ☒ | For null hypothesis testing, the test statistic (e.g. *F*, *t*, *r*) with confidence intervals, effect sizes, degrees of freedom and *P* value noted *Give P values as exact values whenever suitable.* |
| ☒ | ☐ | For Bayesian analysis, information on the choice of priors and Markov chain Monte Carlo settings |
| ☒ | ☐ | For hierarchical and complex designs, identification of the appropriate level for tests and full reporting of outcomes |
| ☒ | ☐ | Estimates of effect sizes (e.g. Cohen's *d*, Pearson's *r*), indicating how they were calculated |

*Our web collection on statistics for biologists contains articles on many of the points above.*

## Software and code

Policy information about availability of computer code

| | |
|---|---|
| Data collection | ORFFinder Python (version 1.8), tblastn and pblast (BLAST suite version 2.13.0+, default parameters) were used to identify inner kinetochore proteins. PCAn and all other custom code related to this project can be found on GitHub: https://github.com/JHelsen/point-centromere-detection. |
| Data analysis | PCAn and all other custom code related to this project can be found on GitHub: https://github.com/JHelsen/point-centromere-detection. This includes code used for phylogenetic analyses and centromere transition simulations. For PCAn, FIMO from the MEME suite v4.11.2, ORFFinder Python v1.8, and a local version of blastp v2.13.0+ were used. For the protein evolutionary analyses, MAFFT v7.505, PAL2NAL v14, Gblocks v0.91b, aBSREL v2.5 and contrast-FEL v0.5 were used. For Alphafold2 predictions and protein structure representations, Colabfold 1.5.5, AlphaFold2-multimer and UCSF ChimeraX v1.8 were used. |

For manuscripts utilizing custom algorithms or software that are central to the research but not yet described in published literature, software must be made available to editors and reviewers. We strongly encourage code deposition in a community repository (e.g. GitHub). See the Nature Portfolio guidelines for submitting code & software for further information.

## Data

Policy information about availability of data

 All manuscripts must include a data availability statement. This statement should provide the following information, where applicable:

- Accession codes, unique identifiers, or web links for publicly available datasets
- A description of any restrictions on data availability
- For clinical datasets or third party data, please ensure that the statement adheres to our policy

All data accompanying this manuscript can be found on FigShare: https://doi.org/10.6084/m9.figshare.c.7630151 (ref 72).
Genomes were downloaded from NCBI (https://www.ncbi.nlm.nih.gov/datasets/genome/), JGI (https://jgi.doe.gov), and from the supplemental data of four studies
(References 48-51: doi: 10.1038/s41586-018-0030-5, doi: 10.1111/mec.13341, doi: 10.1016/j.gene.2024.148722, doi:10.1007/s00253-024-13267-3). Tables with all
genome accession numbers (165 different species + 2737 S. cerevisiae isolates) can be found in the Supplementary Information and on FigShare: https://
doi.org/10.6084/m9.figshare.c.7630151.
The TimeTree database (https://timetree.org) was used to retrieve the divergence time between S. cerevisiae and K. lactis.
The Yeast Gene Order Browser (http://ygob.ucd.ie) was used for synteny checks.
All materials are available upon request.

## Research involving human participants, their data, or biological material

Policy information about studies with human participants or human data. See also policy information about sex, gender (identity/presentation), and sexual orientation and race, ethnicity and racism.

| | |
|---|---|
| Reporting on sex and gender | NA |
| Reporting on race, ethnicity, or other socially relevant groupings | NA |
| Population characteristics | NA |
| Recruitment | NA |
| Ethics oversight | NA |

Note that full information on the approval of the study protocol must also be provided in the manuscript.

# Field-specific reporting

Please select the one below that is the best fit for your research. If you are not sure, read the appropriate sections before making your selection.

☒ Life sciences          ☐ Behavioural & social sciences          ☐ Ecological, evolutionary & environmental sciences

For a reference copy of the document with all sections, see nature.com/documents/nr-reporting-summary-flat.pdf

# Life sciences study design

All studies must disclose on these points even when the disclosure is negative.

| | |
|---|---|
| Sample size | No sample-size calculations were performed. Every Saccharomycetaceae species with available genome assembly and every available Saccharomyces cerevisiae genome assembly were included in the analysis. |
| Data exclusions | For the Saccharomyces cerevisiae intraspecific analyses, some strains were excluded from the main analysis: we removed duplicate strain backgrounds and limited our selection to 1,493 unique strains with phylogenetic information for the main figure. We did however annotate centromeres in the full set of strains (2,737) and the overall frequency of variant centromeres was very similar between the reduced and full dataset. No data was excluded from the plasmid loss experiments. |
| Replication | Plasmid loss assays were set up with at least 12 independent replicate populations. For each experiment, n is reported in the figure legend. The reproducibility of these experiments was very high. |
| Randomization | For population-level experiments, such as plasmid loss assays, randomization is not relevant as yeast cultures are uniform on a population level. All other analyses in this manuscript are computational and did not involve allocating samples into experimental groups. |
| Blinding | Most of the data were analyzed using semi-automated methods. During the experiments, blinding was not possible as each experiment was performed by an individual investigator who was aware of the experimental groups and treatments. |

# Reporting for specific materials, systems and methods

We require information from authors about some types of materials, experimental systems and methods used in many studies. Here, indicate whether each material, system or method listed is relevant to your study. If you are not sure if a list item applies to your research, read the appropriate section before selecting a response.

## Materials & experimental systems

| n/a | Involved in the study |
|-----|----------------------|
| ☒ ☐ | Antibodies |
| ☒ ☐ | Eukaryotic cell lines |
| ☒ ☐ | Palaeontology and archaeology |
| ☒ ☐ | Animals and other organisms |
| ☒ ☐ | Clinical data |
| ☒ ☐ | Dual use research of concern |
| ☒ ☐ | Plants |

## Methods

| n/a | Involved in the study |
|-----|----------------------|
| ☒ ☐ | ChIP-seq |
| ☒ ☐ | Flow cytometry |
| ☒ ☐ | MRI-based neuroimaging |

## Plants

| Seed stocks | NA |
|-------------|-----|
| Novel plant genotypes | NA |
| Authentication | NA |

