## [Peer Review File · Nature]

Progressive coevolution of the yeast centromere and kinetochore

Corresponding Author: Dr Gautam Dey

Version 0:

Reviewer comments:

Referee #1

(Remarks to the Author)

Centromeres are crucial for chromosome segregation, yet they evolve rapidly despite their conserved function. Unlike protein-coding genes, centromeres must either adapt or be newly acquired on each chromosome, making their evolution distinct. This study investigates centromere evolution in 138 budding yeast species and 2,500 strains, demonstrating that new centromeres emerge through genetic drift and are later refined by selection. The kinetochore, which evolves more slowly, plays a key role in determining which centromere variants persist. These findings offer valuable insights into how centromeres evolve while maintaining their essential role in chromosome segregation.

Overall, this is a well-conducted study. The dataset was gathered, generated and analyzed using state-of-the-art methods, and the analyses are robust. The figures are clear and visually appealing, and the results are likely to be of interest to a broad audience. However, while the results are interesting, they do not appear to be particularly unexpected or surprising. Furthermore, in its current form, the manuscript would benefit from additional information and additional arguments to strengthen the conclusions. Finally, it is not yet clear how these results translate to other systems, where centromeres are significantly more complex.

1. As mentioned by the authors, centromeres are typically regions of several hundred kb, made of highly repetitive satellite DNA. The presence of point centromeres, and therefore small structures, is therefore certainly interesting, but it also raises the question of how the authors can relate their findings to these large structures. Anecdotally, the authors should add 'Centromeres evolve progressively through selection at the kinetochore interface IN YEAST' to the title. Or even maybe *Saccharomycetaceae* yeast?

2. Most centromeres are epigenetically defined regions of several hundred kilobases. Centromere identity and function are thought to be determined primarily by epigenetic factors rather than sequence-specific factors for most eukaryotes, including *Arabidopsis thaliana* and many other plants and animals. Do the authors consider that no epigenetic control is necessary for centromere function in yeasts? Regardless of species? And this is why centromeres are compact and point centromeres?

3. After defining a large number of centromeres in yeast, the authors state very clearly that centromeres show extensive diversity across clade. This can indeed be observed visually in Figure 1 but is it possible to have a measurement and a comparison with other regions of the genome?

4. The authors also mentioned that the generated centromere atlas highlighted centromere transitions repeatedly. But how are 'centromere transitions' defined? What are the criteria?

5. There is a limitation to this study, although I can understand that it sometimes needs to be limited. But the study of the evolution and transitions of centromeric structure (i.e. "the evolutionary dynamics underlying centromeric transitions" as mentioned) is limited to a single element of the point centromere, namely CDEII and clearly over the length of this element. I haven't looked closely at the different structures, but is the evolution of the three parts independent or is there a parallel evolution?

6. The authors observed a variation of the CDEII length across species and within a species. Why is that unexpected? I think this is only true if the centromere identity and function is determined and only determined by CDEII and not the other

elements. But I doubt that this is the case.

7. The author conducted *in silico* centromere transition experiments (p. 5 lines 35). Here it would be good for the reader to have some elements in the main text to give him some elements of the simulations carried out.

8. Regarding the hypothesis of tolerance, evolution, and the 10 bp jumps in CDEII, the results appear inconsistent. If CDEII length variations of 10 bp are truly tolerated, we should observe both +10 bp and -10 bp changes being at the same level. Indeed, since increase or decrease is relative in this context, one would expect symmetric retention patterns. Yet, in Figure 4C, the relative retention for -10 bp is similar to that of -5 bp and +5 bp, contradicting the hypothesis. Therefore, the proposed model does not fully align with the observed data.

9. The author discusses the adaptation of the kinetochore machinery, but adaptation to what exactly? Furthermore, wouldn't the Cbf3 complex also be subject to episodic diversifying selection? Did the author find any evidence supporting this idea?

Referee #2

(Remarks to the Author)

Summary:

Helsen et al describe the evolution of point centromeres in budding yeast species. The authors developed and used a tool to annotate centromeres in yeasts and found that even though the centromeres in budding yeasts are genetically defined and tend to be on the smaller side (up to 200 bp), there is extensive variation across budding yeasts. The authors also found that the changes in the centromeric DNA are happening both through drift and selection. In addition to this, one of their very exciting discoveries is that the inner kinetochore proteins are (slowly) 'compensating' for the changes in the centromere sequence. Lastly, the authors show that strains/species can 'gain' new centromeres from outcrossed diploids (where the parents have different centromeric variants) undergoing meiosis. These results open new questions in the centromere evolution field as to how some kinetochore proteins can be promiscuous and interact with two distinct centromeres or how the sequence in such proteins can constrain the overall evolution of the centromeric DNA. Exciting time to be in the field!! It is a very well written paper with clear and visually pleasing figures.

Major revisions:

1. Please mention in the text that while budding yeasts are generally diploid, the modeling shown in Figure 2e is done in haploids. The authors could also add the analysis in diploids which will likely look very similar and then it's done in the proper ploidy of these organisms.
2. Plasmid Loss Assay shown in Figure 2h, Figure 4c and Methods.
 - a. I would call it plasmid retention assay as that makes more sense as to what is assayed and reported in the figures.
 - b. To both the figures and the methods, please add the approximate number of generations that the cells are undergoing after removal of selection (-G418)
 - c. Please add the following controls: to figure 2- add the *S cerevisiae* (BY4741) CEN plasmid as control. Also, the authors need to account for the rate of plasmid loss in an empty plasmid (e.g., without a CEN, a scrambled CEN sequence, or both).
 - d. The fluorescent protein is often maintained for a few divisions after plasmid loss. The authors need to determine the rate of false positives unless they used a degron in their mNeogreen (but it isn't reported in the methods). They could also check for the presence/absence of the plasmid by plating sorted mNeogreen positive cells onto YPD and YPD + G418 plates.
 - e. Please describe in the methods how the plasmids were made, and the sequences of the oligos used (or if they were synthesized).
 - f. Add to the methods how the analysis of the flow cytometry data was done, what gating was used, and how many replicates were done per strain. Also, specify if the normalized median fluorescence is what is plotted in the graphs. Did the authors use an additional cytoplasmic fluorescent protein inserted in the genome to normalize by cell volume? or do they normalize by SSC as a proxy for volume?
 - g. Explain in methods how the authors confirmed that the *J. jinghongensis* species underwent the same number of generations as the other species given that this species was grown at a different temperature and the authors are comparing between them.
 - h. For both graphs in the figures, make sure to do a statistical comparison to determine if the observed plasmid retention/loss is significant.
3. Figure 4c, explain in the methods how the CEN variants were made. How was the sequence length determined? For example, when removing 5 bp was that from the 5' or 3' of the sequence? Also, to strengthen the argument that the kinetochore interface may interact with a given length and the importance of that, the authors could insert ~10bp of scrambled DNA sequence.
4. Add a strain table, oligo table, and plasmid table describing all strains used and their construction.

Minor revisions:

Figure 1- add the name of the strains of the outgroups in the figure legend.

Page 5, line 41- should say fig 2g, not 2f

Page 5, Line 44 – should say Fig 2h, not 2g

Page 8, line 15 and figure 3. Do the authors really mean recombination or just meiosis? Because they are not really observing homologous recombination between centromeres just allele sorting.

Page 8, line 1. Did the authors look if SPO11 oligos are found near centromeres

(<https://pmc.ncbi.nlm.nih.gov/articles/PMC3063416/>)? Another hypothesis is that during meiosis, DSBs that may happen near CENs repair with the sister chromatid and expand.

Page 10- Line 16-18, "The ability to tolerate such jumps in centromere length seems to be a defining feature of the point centromere clade (Fig. 4a)". Not sure I understand what the authors mean by this, "defining" may seem like a strong word given that only a handful of species exhibit this in Figure 4a. If I understood this wrong, then it would be useful to expand this a little better.

Page 9- line 25, and Page 16, AF: due to the new release of AF3, the authors may want to re model their data where now they can add both Cbf1 and the CDEI motif and see how they fit together, especially for the mutants where the motif length varies.

Page 10, line 15: Maybe instead of "mitotic efficiency" use "ability to properly segregate plasmids" which is more specific.

Page 11, line 28. Just curious, but did the authors tried to test their plasmid loss assay in the *Jamesozyma* strains under different growing conditions?

Page 11, line 31-33, the authors may want to add 'despite their potential fitness costs' at the end of that sentence, as the idea of centromere drive is based on selfishness.

Page 11, Line 34-35, This is correct, however, the work by the Zanders lab on how meiotic drivers shape the evolution of (yeast, male) meiosis could explain your observations. <https://elifesciences.org/articles/57936>

Page 14, line 40 – as mentioned above, I believe the authors are using 'recombination' wrong. I think they mean 'meiosis'.

Page 15, line 20 – reference protocol even if it is 'standard'

Page 15, line 22 – add recipe of YPAD

Extended Data Fig 1 is unreadable.

Extended Data Fig 6. There is a missing 's' in *Zygosaccharomyces*

(Remarks on code availability)

The code is available and the README file provides good instructions to reproduce their data.

Referee #3

(Remarks to the Author)

This manuscript develops a computational method to identify centromere sequences in yeast closely related to the model yeast *Saccharomyces cerevisiae*. The authors then compare these sequences across these species to examine variation both between and within species. They focus on variation in the central element of the centromere, CDEII which, in *S. cerevisiae* wraps the centromeric nucleosome. Surprisingly, the authors identify variation in the length of CDEI, primarily between species. Simulations infer that CDEII variants change slowly by drift and selection and spread through populations by sex. Finally, the authors examine the Cbf1 protein which is a sequence-specific DNA binding protein that is both at centromeres and also some promoters, and is unique to "point" centromeres. They find that Cbf1 does find some diversifying selection in its N terminal tail. The authors speculate that this affects the interaction with the kinetochore interface, but this is yet to be tested. Overall, this manuscript makes some interesting observations and raises some interesting hypotheses. However, the conclusions drawn are over-stated based on the data shown, the generality of the findings for centromeres in other systems seems unlikely and some of the conclusions are premature without testing experimentally *in vivo*.

1. The authors have not demonstrated that "centromeres evolve progressively through selection at the kinetochore interface". The only investigation related to this was the examination of the diversity of Cbf1, but they have not shown that any of these changes affect the "kinetochore interface".

2. Conceptually, evolution of genetically defined centromeres is very different to evolution of epigenetically defined centromeres. In fact, it could be argued that evolution of genetically defined centromeres has more in common with evolution of genes (or at least gene expression) than with epigenetic centromeres. Both gene expression and point centromeres can be defined by site specific DNA binding proteins, while this is not the case for regional centromeres, where other mechanisms are at play. Therefore, while the study is interesting from the point of view of yeast biology, it seems unlikely that the findings can be extrapolated to more complex centromeres. There needs to be greater transparency on this point.

3. Centromeres are defined *in silico* by sequence features, but whether these sequences actually act as centromeres *in vivo* is not known. To confirm the method of centromere identification it would be necessary to provide some *in vivo* evidence, for example ChIP-Seq of a kinetochore protein or CENPA.

4. Similarly, how confident can the authors be in the sequence assemblies, particularly when considering the CDEII lengths? Have the authors carried out directed sequencing to confirm?

5. Figure 3f. This figure and the conclusion that the variation comes from microhomology mediated recombination is quite difficult to follow. This needs better explanation.

6. Figure 4c. Can the authors explain why CDEII that is 10 bp shorter does not support plasmids stability? Centromeres with a shorter length were identified in Figure 1. How could these work?

7. The manuscript would be strengthened by further experimental testing of the assertions/hypotheses from the *in silico* data. The authors could, for example, build hybrid centromeres and use exogenous expression to determine whether Cbf1 variants could support different CDEII lengths in a heterologous yeast (e.g. in *Saccharomyces cerevisiae*).

Version 1:

Reviewer comments:

Referee #1

(Remarks to the Author)

Although the authors provided additional analyses and clarifications in response to previous comments, several of my concerns were only partially addressed. For example, while the inclusion of new analyses in Mucoromycota and additional experimental controls is welcome, some key questions remain. In particular, I remain concerned that, although the results are interesting, they are not particularly surprising in the organisms studied. The manuscript would benefit from more robust information and arguments to support the conclusions, and the extent to which these results can be generalized to systems with more complex centromeres remains unclear. Overall, the study offers intriguing observations and raises interesting hypotheses. However, in its current form, some conclusions appear overstated relative to the data, the generality of the results beyond fungi is questionable, and several claims remain premature without further validation.

Referee #2

(Remarks to the Author)

Helsen et al addressed all my previous comments and their paper is better than their previous submission. The paper is clear and the additions to their current version have made it even more exciting.

Few suggestions:

1. In page 6, starting on line 6, we suggest authors to add "centromere LIKELY change one by one".
2. Figure 3d, we suggest authors show all homozygous chromosome in CBS2271 as it makes it confusing if we are looking at haploids or homozygous diploids.

Referee #3

(Remarks to the Author)

The authors have satisfactorily addressed the points raised by the reviewers. In particular, inclusion of the new data on Mucoromycota has validated the generality of their approach. The authors have also included textual changes comparing the features of point and regional centromeres which is appreciated.

The authors additionally provide some experimental data that supports the idea that Cbf1 is adapted to Cdell centromere length. This is also appreciated, however, the plasmid retention assay is quite indirect. The conclusion that the phenomenon observed is due to selection "at the kinetochore interface" is premature and at this point rather speculative. Therefore, the title of the manuscript should be toned down to reflect this. On the other hand, the authors have robustly shown that centromeres evolve progressively. The mechanisms that provide the selection to constrain centromere length has not been fully demonstrated in this manuscript.

Referee #4

(Remarks to the Author)

I co-reviewed this manuscript with one of the reviewers who provided the listed reports.

Referee #1 (Remarks to the Author):

Centromeres are crucial for chromosome segregation, yet they evolve rapidly despite their conserved function. Unlike protein-coding genes, centromeres must either adapt or be newly acquired on each chromosome, making their evolution distinct. This study investigates centromere evolution in 138 budding yeast species and 2,500 strains, demonstrating that new centromeres emerge through genetic drift and are later refined by selection. The kinetochore, which evolves more slowly, plays a key role in determining which centromere variants persist. These findings offer valuable insights into how centromeres evolve while maintaining their essential role in chromosome segregation.

Overall, this is a well-conducted study. The dataset was gathered, generated and analyzed using state-of-the-art methods, and the analyses are robust. The figures are clear and visually appealing, and the results are likely to be of interest to a broad audience. However, while the results are interesting, they do not appear to be particularly unexpected or surprising. Furthermore, in its current form, the manuscript would benefit from additional information and additional arguments to strengthen the conclusions. Finally, it is not yet clear how these results translate to other systems, where centromeres are significantly more complex.

We thank the reviewer for their helpful comments, and believe that by addressing them our manuscript has become more clear and nuanced. Specifically, we:

- Analyse more complex centromeres in a clade ~675 million years removed from the Saccharomycetaceae, and use it to strengthen our conclusions (see details below in our response to individual comments)
- Do a better job positioning the yeast centromere within eukaryotic centromere diversity
- Provide additional information, arguments and experiments to further support our conclusions (see details below in our response to individual comments)

With regards to the results not appearing to be unexpected or surprising, we would say that our work does indeed describe some of the most basic evolutionary principles hypothesized to underlie centromere evolution. However, through our study, we can now for the first time confirm these hypotheses using real evolutionary data of tens of thousands of centromeres and *in vivo* centromere function experiments.

1. As mentioned by the authors, centromeres are typically regions of several hundred kb, made of highly repetitive satellite DNA. The presence of point centromeres, and therefore small structures, is therefore certainly interesting, but it also raises the question of how the authors can relate their findings to these large structures. Anecdotally, the authors should add 'Centromeres evolve progressively through selection at the kinetochore interface IN YEAST' to the title. Or even maybe Saccharomycetaceae yeast?

To evaluate whether our approach and findings can be applied outside of yeast and to more complex centromeres, our revised manuscript now contains analyses for a new eukaryotic clade: the Mucoromycota, a lineage of basal fungi with more complex mosaic centromeres about 675 million years removed from the Saccharomycetaceae:

Mucoromycota centromeres are 15-73 kb long, and consist of a small (300-400 bp) core region which is bound by the kinetochore protein, flanked by retrotransposons on either side. The core region contains a DNA motif and AT-rich region reminiscent of the Saccharomycetaceae point centromere.

Using an adapted version of our pipeline, we annotated centromeres for 8 Mucoromycota, and observed evolutionary trends very similar to those we see in the Saccharomycetaceae. The data were used in Figures 1, 2, and 3 to strengthen our conclusions; we see that:

- Similar to the Saccharomycetaceae, within a single Mucor genome, the length of the AT-rich region is remarkably consistent, yet it can vary by nearly two-fold between different species (Fig. 1)
- Similar to the Saccharomycetaceae, Mucoromycota can have genomes with 'mixed' types of centromeres (Fig. 2)
- Similar to the Saccharomycetaceae, Mucoromycota can combine centromere variants through sex (Fig. 3)

Many 'complex' centromeres still contain motif-like elements that are bound by specific kinetochore proteins (e.g. CENP-B binds the CENP-B-box motif in mammalian centromeric satellites and CENP-B like elements are found even outside of mammals). Motifs are also found in several other fungi with 'regional' centromeres and in certain diatom centromeres. Given that, we expect that, as the quality and density of genome assemblies continues to explode, our predictions could soon be experimentally validated in clades with complex centromeres in other eukaryotic clades. We now also discuss this in the discussion.

We have also attempted to better position the yeast centromere within eukaryotic centromere diversity. The manuscript now contains an overview of the size distribution of eukaryotic centromeres. This overview also highlights that small centromeres are neither rare nor unique to fungi.

2. Most centromeres are epigenetically defined regions of several hundred kilobases. Centromere identity and function are thought to be determined primarily by epigenetic factors rather than sequence-specific factors for most eukaryotes, including *Arabidopsis thaliana* and many other plants and animals. Do the authors consider that no epigenetic control is necessary for centromere function in yeasts? Regardless of species? And this is why centromeres are compact and point centromeres?

Centromeres of many species are smaller (see previous comment), but many do indeed contain both epigenetic and genetic elements. Saccharomycetaceae do not have the machinery necessary for DNA methylation and it has been proposed that this could indeed have led to a reduction in

centromere size. While the literature suggests that epigenetic control does not play a major role for centromere function in Saccharomycetaceae, we believe that meaningful conclusions can be drawn about the general evolutionary dynamics of centromeres, especially since we now include data from another clade with more complex centromeres (see previous comment). Additionally, understanding how stretches of centromeric DNA coevolve with the kinetochore machinery is important even in centromeres with more epigenetic control.

3. After defining a large number of centromeres in yeast, the authors state very clearly that centromeres show extensive diversity across the clade. This can indeed be observed visually in Figure 1 but is it possible to have a measurement and a comparison with other regions of the genome?

Since these regions evolve extremely fast, syntenic centromeres have little to no sequence similarity between species and don't align in any meaningful way, precluding us from calculating mutation rates. Within species, they can be aligned, and for *S. paradoxus* and *S. cerevisiae* two papers have already shown that centromeres (and the *CDEII* region especially) are the fastest evolving regions in the genome (Bensasson et al. 2008, *Genetics*; Bensasson 2011, *BMC Ecology and Evolution*). We use Fig. 1 and Extended Data Fig. 1 to highlight differences between species, rather than specifically quantifying this variation.

4. The authors also mentioned that the generated centromere atlas highlighted centromere transitions repeatedly. But how are 'centromere transitions' defined? What are the criteria?

Indeed, a centromere 'transition' can be defined in different ways. In this case, we chose to define a transition as going from one particular *CDEII* length to another, as the binary nature of *CDEII* length variation makes it the ideal feature to highlight different aspects of the evolutionary dynamics of centromeres. For the transitions highlighted in the manuscript (*Jamesozyma*, *S. cerevisiae* isolates, *Vanderwaltozyma*, ...), other parts of the centromere (*CDEI*, *CDEIII*) didn't show major differences between species/strains (see Extended Data Fig. 1), allowing us to examine the effect of one centromere feature at the time.

5. There is a limitation to this study, although I can understand that it sometimes needs to be limited. But the study of the evolution and transitions of centromeric structure (i.e. "the evolutionary dynamics underlying centromeric transitions" as mentioned) is limited to a single element of the point centromere, namely *CDEII* and clearly over the length of this element. I haven't looked closely at the different structures, but is the evolution of the three parts independent or is there a parallel evolution?

For the reasons stated above, we chose to mainly focus on *CDEII* length, but there are indeed also instances where the other two parts of the centromere show clear signs of evolution. We highlighted two examples in the text: e.g., *Huiozyma* spp. lack the conserved CCGAA motif, while part of the *CDEIII* motif is more conserved in *Sungouiozyma* spp. In these two examples, evolution of the *CDEIII* motif seems to be independent of *CDEII* length evolution.

6. The authors observed a variation of the *CDEII* length across species and within a species. Why is that unexpected? I think this is only true if the centromere identity and function is determined and only determined by *CDEII* and not the other elements. But I doubt that this is the case.

While it is maybe not unexpected that *CDEII* length would evolve, what is surprising are the clear patterns we observe in relation to the evolution of this feature. For most species, there is very little inter-species variation; variation we do see on shorter evolutionary distances follows clear rules (10 base pair differences), and on larger evolutionary distances the length can almost quadruple in size. For the canonical *S. cerevisiae* point centromere, a *CDEII* length of 85 was shown to allow for one wrap around the centromeric nucleosome. It is therefore quite unexpected to find such a wide variety of *CDEII* lengths across the clade.

7. The author conducted *in silico* centromere transition experiments (p. 5 lines 35). Here it would be good for the reader to have some elements in the main text to give him some elements of the simulations carried out.

We added the following description to the main text:

“Starting with an individual carrying 16 'A' type centromeres, the simulation iteratively transitions a single, randomly selected centromere to type 'B' or not, based on a retention probability.”

To not add too many technical details to the main text, we opted to provide more detailed information in both the figure legends and the methods section.

8. Regarding the hypothesis of tolerance, evolution, and the 10 bp jumps in *CDEII*, the results appear inconsistent. If *CDEII* length variations of 10 bp are truly tolerated, we should observe both +10 bp and -10 bp changes being at the same level. Indeed, since increase or decrease is relative in this context, one would expect symmetric retention patterns. Yet, in Figure 4C, the relative retention for -10 bp is similar to that of -5 bp and +5 bp, contradicting the hypothesis. Therefore, the proposed model does not fully align with the observed data.

The vast majority of species show 1 short and 1 long *CDEII* in cases where they have mixed *CDEII* lengths. The collapsed tree does indeed make it look like there is a +/- symmetry, but this is only true on a genus level. For example: in a genus with a median length of 80 bp, there can be species with 70 bp and 80 bp variants and *other* species with 80 bp and 90 bp variants. To make this more clear, we added an uncollapsed version of the tree in Fig. 4 as Extended Data Fig. 6. The trend is especially obvious in the *Lachancea* and *Vanderwaltozyma* genus:

In the case of *S. cerevisiae*, most *CDEII* length variants are longer (see variant frequency Fig. 3c). We hypothesize that a species' kinetochore structure will determine whether a -10 or +10 bp variant is preferred, and for *S. cerevisiae*, based on what we see in nature, it seems like longer variants are tolerated, and this is also what we observe in our experiments.

We now also mention this in the main text:

“Similar to what we observed for natural isolates (Fig. 3), *S. cerevisiae* seems to tolerate longer centromere variants better than shorter variants.”

9. The author discusses the adaptation of the kinetochore machinery, but adaptation to what exactly? Furthermore, wouldn't the Cbf3 complex also be subject to episodic diversifying selection? Did the author find any evidence supporting this idea?

We hypothesize that initially, new centromere variants can arise and spread through populations through drift as long as they maintain compatibility with the current kinetochore machinery. However, we also observe that new variants can become more efficient at segregating than 'old' variants (e.g., *J. spencerorum* - Fig. 2), implying that at a certain point, something in the kinetochore must have

changed. The exact evolutionary pressures and sequence of events underlying this change remain unclear, but one (of several) scenarios one could come up with is that a change in niche (e.g., temperature, which is known to alter microtubule dynamics) could have altered the relative segregation efficiency of different variants, generating pressure to become better at segregating one particular variant.

Within the genus we focussed on to test for episodic diversifying selection (*Jamesozyma* spp.), *CDEIII* motifs are relatively similar between species and we did test for, but did not observe signatures of selection in the Cbf3 complex. We also specifically looked for signs of positive selection in the Cbf3 complex across a clade which contains different *CDEIII* motifs (the clade spanning from *Kazachstania martiniae* to *Arxiozyma spVP05*), but could not find any significant signature correlating with changes in *CDEIII*. In general, these types of analyses work better on smaller evolutionary distances (within-genus), which could be one of the reasons why we did not pick anything up in this clade.

Referee #2 (Remarks to the Author):

Summary:

Helsen et al describe the evolution of point centromeres in budding yeast species. The authors developed and used a tool to annotate centromeres in yeasts and found that even though the centromeres in budding yeasts are genetically defined and tend to be on the smaller side (up to 200 bp), there is extensive variation across budding yeasts. The authors also found that the changes in the centromeric DNA are happening both through drift and selection. In addition to this, one of their very exciting discoveries is that the inner kinetochore proteins are (slowly) 'compensating' for the changes in the centromere sequence. Lastly, the authors show that strains/species can 'gain' new centromeres from outcrossed diploids (where the parents have different centromeric variants) undergoing meiosis. These results open new questions in the centromere evolution field as to how some kinetochore proteins can be promiscuous and interact with two distinct centromeres or how the sequence in such proteins can constrain the overall evolution of the centromeric DNA. Exciting time to be in the field!! It is a very well written paper with clear and visually pleasing figures.

We thank the reviewer for their insightful comments, as we believe addressing them improved our manuscript significantly.

Major revisions:

1. Please mention in the text that while budding yeasts are generally diploid, the modeling shown in Figure 2e is done in haploids. The authors could also add the analysis in diploids which will likely look very similar and then it's done in the proper ploidy of these organisms.

We now include the same analyses for diploids in Extended Data Fig. 4 c and d:

The analysis does indeed look very similar, although, as expected, more transitions are necessary to achieve full transitions.

In the text, we now say:

“These simulations show that full centromere transitions are indeed not possible without selection for the new variant, both in haploids and diploids.”

We decided to keep the simulations for haploids in the main figure, because they are slightly easier to interpret. A large fraction of the species in our dataset with known ploidy are indeed diploid, but some clades also contain natural haploids (e.g. *Vanderwaltozyma polyspora*, *Nakaseomyces* spp., *Kluyveromyces lactis*, *Emmentothecium gossypii*, many *Lachancea* spp., ...).

2. Plasmid Loss Assay shown in Figure 2h, Figure 4c and Methods.

a. I would call it plasmid retention assay as that makes more sense as to what is assayed and reported in the figures.

We agree, this is a better framing. We replaced all instances of “plasmid loss” by “plasmid retention”.

b. To both the figures and the methods, please add the approximate number of generations that the cells are undergoing after removal of selection (-G418)

Since we inoculate from a saturated culture and let cells grow to saturation after removal of selection, the number of generations is mostly determined by the dilution factor: 1:256 ~ 8, plus 1.5-2.5 generations from the initial dilution to $OD_{600} = 1$ makes ~10 generations after removal of selection. We added this information to the figures, figure legends and methods.

c. Please add the following controls: to figure 2- add the *S. cerevisiae* (BY4741) CEN plasmid as control. Also, the authors need to account for the rate of plasmid loss in an empty plasmid (e.g., without a CEN, a scrambled CEN sequence, or both).

Fig. 2i now contains retention assays with empty plasmid controls and *S. cerevisiae* controls. It actually really helped us highlight the functionality of the centromere sequences, so thank you for this suggestion!

d. The fluorescent protein is often maintained for a few divisions after plasmid loss. The authors need to determine the rate of false positives unless they used a degron in their mNeogreen (but it isn't reported in the methods). They could also check for the presence/absence of the plasmid by plating sorted mNeogreen positive cells onto YPD and YPD + G418 plates.

We determined the false positive rates by sorting mNeogreen positive single cells (using a BD FACSDiscover S8 image sorter) and plating them on YPD and YPD + G418 plates. The results are shown in Extended Data Fig. 4e:

The false positive rate is negligible for *S. cerevisiae*, but about 27-37% for the different *Jamesozyma* species. Because of this, but also because of inevitable differences in growth dynamics between the species, we only compare retention rates within the same species. Relative retention rates are always normalized within one species and never across species.

Note on the lower colony counts for *J. spencerorum*: this species seems to somewhat stick to plastic, so we suspect many cells were stuck to the tube instead of being plated.

e. Please describe in the methods how the plasmids were made, and the sequences of the oligos used (or if they were synthesized).

Details on how the plasmids were made can now be found in the methods section and Supplementary Information (Data S7 | Cloning strategy for plasmid construction):

“Plasmids were made using NEBuilder HiFi assembly (NEB) or Q5 site-directed mutagenesis (NEB) and verified by whole plasmid sequencing (Plasmidsaurus). Oligos and cloning strategies for each plasmid can be found in the Supplementary information.”

All the plasmid maps are also available on Figshare.

f. Add to the methods how the analysis of the flow cytometry data was done, what gating was used, and how many replicates were done per strain. Also, specify if the normalized median fluorescence is what is plotted in the graphs. Did the authors used an additional cytoplasmic fluorescent protein inserted in the genome to normalize by cell volume? or do they normalized by SSC as a proxy for volume?

The methods section now reads:

“...the proportion of fluorescent cells was measured by flow cytometry on an Accuri C6 (BD Biosciences), or a FACSymphony A3 (BD Biosciences) (minimum 30,000 singlets). Forward and side scatter were used to select singlets, and the fluorescence distribution of the T0 sample was used to set the gate distinguishing fluorescent from non-fluorescent cells. Using those proportions, the relative mNeogreen retention was determined by performing within-species normalization, by dividing each datapoint by the mean of the lowest performing plasmid within the same species and experiment (Fig. 2) or by dividing by the mean of the wild type (Fig. 4).”

The number of replicates for each experiment is mentioned in the figure legends.

What is plotted in all graphs is the relative mNeogreen retention. It is a per-species normalization (see methods text above), since comparisons of plasmid retention between species are not fair (differences in false positive rate, differences in growth dynamics). We did not normalize for the cell volume but did not observe differences in the forward and side scatter distributions for the selected singlet populations between samples from the same species.

We opted to normalize to the lowest performing plasmid in Fig. 2 to highlight how plasmids with a centromere outperform those a centromere sequence. In Fig. 4, we normalized to the *S. cerevisiae* wild type.

g. Explain in methods how the authors confirmed that the *J. jinghongensis* species underwent the same number of generations as the other species given that this species was grown at a different temperature and the authors are comparing between them.

As mentioned above, for all the species we inoculate from a saturated culture and let cells grow to saturation after removal of selection so that the number of generations is mostly determined by the dilution factor. We also chose to only compare centromere performance within species, not between species (see above).

The reason *J. jinghongensis* was grown at a different temperature is that it does not grow at 30°C. On the other hand, *J. spencerorum* shows impaired growth at 25°C:

h. For both graphs in the figures, make sure to do a statistical comparison to determine if the observed plasmid retention/loss is significant.

We added statistical comparisons to all relevant comparisons in Fig. 2 and Fig. 4.

3. Figure 4c, explain in the methods how the CEN variants were made. How was the sequence length determined? For example, when removing 5 bp was that from the 5' or 3' of the sequence? Also, to strengthen the argument that the kinetochore interface may interact with a given length and the importance of that, the authors could insert ~10bp of scrambled DNA sequence.

Details on how plasmids with CEN variants were made are now part of the methods section and Supplementary Information (Data S7 | Cloning strategy for plasmid construction).

The length of the sequence was verified by whole plasmid sequencing (Plasmidsaurus).

Base pairs were removed closer to the 5' end and bases were added near the middle. In general, we tried to maintain a similar average AT% for the *CDEII* sequence. As we generated our new plasmids by site-directed mutagenesis, some variants were generated by accident (details in Supplementary Information), but for every plasmid, the full plasmid sequence was verified so that the only variation present was located within the *CDEII* region. An alignment of the centromere variants can now be found in Extended Data Fig. 7a:

4. Add a strain table, oligo table, and plasmid table describing all strains used and their construction.

We added a strain list (Supplementary information - Data S8), an oligo list (Supplementary information - Data S9), and all plasmids used in this study together with details on their construction (Supplementary information - Data S7). Plasmid sequences are also available on FigShare.

Minor revisions:

Figure 1- add the name of the strains of the outgroups in the figure legend.

The names of the outgroup species were added to the legend of Fig. 1 and Extended Data Fig. 1.

Page 5, line 41- should say fig 2g, not 2f

We corrected this.

Page 5, Line 44 – should say Fig 2h, not 2g

We corrected this.

Page 8, line 15 and figure 3. Do the authors really mean recombination or just meiosis? Because they are not really observing homologous recombination between centromeres just allele sorting.

Thank you! We changed it to meiosis.

Page 8, line 1. Did the authors look if SPO11 oligos are found near centromeres (<https://pmc.ncbi.nlm.nih.gov/articles/PMC3063416/>)? Another hypothesis is that during meiosis, DSBs that may happen near CENs repair with the sister chromatid and expand.

We did not look at SPO11 oligos, but added the alternative hypothesis in the main text:

“Alternatively, they could have arisen through repair of double-strand breaks near centromeres during meiosis.”

Page 10- Line 16-18, “The ability to tolerate such jumps in centromere length seems to be a defining feature of the point centromere clade (Fig. 4a)”. Not sure I understand what the authors mean by this, “defining” may seem like a strong word given that only a handful of species exhibit this in Figure 4a. if I understood this wrong, then it would be useful to expand this a little better.

We meant to say that the ability to tolerate these jumps is present across most of the point centromere clade. For some species it is indeed more prevalent, but for others (like the *Saccharomyces* species), we only see it when looking through many genomes. We clarified this in the main text:

“The ability to tolerate such jumps in centromere length seems to be present across the point centromere clade, even though the expression of this trait varies between species (Fig. 4a, Extended Data Fig. 6). While some clades, like *Lachancea* and *Vanderwaltozyma*, show a consistent mix of *CDEII* lengths, others like *Saccharomyces* only show the ~10 base pair jumps in a proportion of the population (Fig. 3).”

Page 9- line 25, and Page 16, AF: due to the new release of AF3, the authors may want to re model their data where now they can add both Cbf1 and the CDEI motif and see how they fit together, especially for the mutants where the motif length varies.

We tried AF3, but it completely failed to correctly recognize the motif, even when we added in additional kinetochore components.

Page 10, line 15: Maybe instead of “mitotic efficiency” use “ability to properly segregate plasmids” which is more specific.

In both the legend for Fig. 2 and Fig. 4, we changed it to:

“the ability of different centromere variants to support plasmid segregation”

Page 11, line 28. Just curious, but did the authors tried to test their plasmid loss assay in the *Jamesozyma* strains under different growing conditions?

We did try our plasmid loss assay at three different temperatures in *J. spencerorum*, but we did not observe a significant difference in relative retention between the different temperatures. This is possibly because the *J. spencerorum* Cbf1 has such a large effect, even if it is put in *S. cerevisiae* (which presumably changes the cellular context even more) (new Fig. 4g).

Page 11, line 31-33, the authors may want to add 'despite their potential fitness costs' at the end of that sentence, as the idea of centromere drive is based on selfishness.
Thank you for this suggestion! We added it at the end of the sentence.

Page 11, Line 34-35, This is correct, however, the work by the Zanders lab on how meiotic drivers shape the evolution of (yeast, male) meiosis could explain your observations. <https://elifesciences.org/articles/57936>
We hadn't thought about that, thank you! We amended the text to read:
"many species in this clade strictly undergo male meiosis where all four meiotic products are viable in the absence of other meiotic drivers"

Page 14, line 40 – as mentioned above, I believe the authors are using 'recombination' wrong. I think they mean 'meiosis'.
We corrected this throughout the text.

Page 15, line 20 – reference protocol even if it is 'standard'
A reference for the protocol was added.

Page 15, line 22 – add recipe of YPAD
We added the recipe:
"YPAD (1% w/v yeast extract (BD), 2% w/v peptone (BD), 2% w/v glucose (Merck), 40 mg/L adenine sulfate (Sigma))"

Extended Data Fig 1 is unreadable.
We provided a higher-resolution image in our new submission. Both as a separate file and in the compiled PDF.

Extended Data Fig 6. There is a missing 's' in *Zygosaccharomyces*
Thank you! We corrected this.

Referee #2 (Remarks on code availability):

The code is available and the README file provides good instructions to reproduce their data.

Thank you!

Referee #3 (Remarks to the Author):

This manuscript develops a computational method to identify centromere sequences in yeast closely related to the model yeast *Saccharomyces cerevisiae*. The authors then compare these sequences across these species to examine variation both between and within species. They focus on variation in the central element of the centromere, CDEII which, in *S. cerevisiae* wraps the centromeric nucleosome. Surprisingly, the authors identify variation in the length of CDEI, primarily between species. Simulations infer that CDEII variants change slowly by drift and selection and spread through populations by sex. Finally, the authors examine the Cbf1 protein which is a sequence-specific DNA binding protein that is both at centromeres and also some promoters, and is unique to “point” centromeres. They find that Cbf1 does find some diversifying selection in its N terminal tail. The authors speculate that this affects the interaction with the kinetochore interface, but this is yet to be tested. Overall, this manuscript makes some interesting observations and raises some interesting hypotheses. However, the conclusions drawn are over-stated based on the data shown, the generality of the findings for centromeres in other systems seems unlikely and some of the conclusions are premature without testing experimentally *in vivo*.

We thank the reviewer for their comments, and believe that by addressing them our manuscript has significantly improved. Specifically, we:

- Provide experimental evidence that evolution of a kinetochore protein can alter which centromere variants are preferentially retained.
- Analyse more complex centromeres in a clade ~675 million years removed from the Saccharomycetaceae, and use it to strengthen our conclusions
- Do a better job positioning the yeast centromere within eukaryotic centromere diversity
- Provide additional information, controls and experiments to further support our conclusions

1. The authors have not demonstrated that “centromeres evolve progressively through selection at the kinetochore interface”. The only investigation related to this was the examination of the diversity of Cbf1, but they have not shown that any of these changes affect the “kinetochore interface”.

A secondary conclusion is indeed that the kinetochore interface could also eventually evolve to compensate for an evolving centromere (Fig. 4d-f), and while we don't have experimental evidence showing how selection acts *on* the kinetochore, we now provide experimental evidence that evolution of a kinetochore protein can indeed alter which centromere variants are preferentially retained:

“To test whether these different Cbf1 variants can indeed alter which centromere variants are preferentially retained, we replaced the endogenous CBF1 gene in *S. cerevisiae* with different *Jamesozyma* variants, and measured the relative retention rate of plasmids with either short or long *Jamesozyma* centromeres. Remarkably, the *J. spencerorum* Cbf1 protein made long *Jamesozyma* centromeres significantly more efficient than short centromeres (Fig. 4g), suggesting that mutations in this protein might indeed explain why the long variant acquired a selective advantage and reached fixation within the genome.”

The title, however, refers to how the structure of the kinetochore acts as a selection filter for centromere evolution, and this is something we demonstrate throughout the manuscript, using the

evolutionary record as well as synthetic experiments to show that *only some centromere mutations are tolerated* and that the kinetochore is an important constraining factor for these mutations (+/-10 bp).

2. Conceptually, evolution of genetically defined centromeres is very different to evolution of epigenetically defined centromeres. In fact, it could be argued that evolution of genetically defined centromeres has more in common with evolution of genes (or at least gene expression) than with epigenetic centromeres. Both gene expression and point centromeres can be defined by site specific DNA binding proteins, while this is not the case for regional centromeres, where other mechanisms are at play. Therefore, while the study is interesting from the point of view of yeast biology, it seems unlikely that the findings can be extrapolated to more complex centromeres. There needs to be greater transparency on this point.

To evaluate whether our approach and findings can be applied outside of yeast and to more complex centromeres, our revised manuscript now contains analyses for a new eukaryotic clade: the Mucoromycota, a lineage of basal fungi with more complex mosaic centromeres about 675 million years removed from the Saccharomycetaceae:

Mucoromycota centromeres are 15-73 kb long, and consist of a small (300-400 bp) core region which is bound by the kinetochore protein, flanked by retrotransposons on either side. The core region contains a DNA motif and AT-rich region reminiscent of the Saccharomycetaceae point centromere.

Using an adapted version of our pipeline, we annotated centromeres for 8 Mucoromycota, and observed evolutionary trends very similar to those we see in the Saccharomycetaceae. The data were used in Figures 1, 2, and 3 to strengthen our conclusions; we see that:

- Similar to the Saccharomycetaceae, within a single Mucor genome, the length of the AT-rich region is remarkably consistent, yet it can vary by nearly two-fold between different species (Fig. 1)
- Similar to the Saccharomycetaceae, Mucoromycota can have genomes with 'mixed' types of centromeres (Fig. 2)
- Similar to the Saccharomycetaceae, Mucoromycota can combine centromere variants through sex (Fig. 3)

Many 'complex' centromeres still contain motif-like elements that are bound by specific kinetochore proteins (e.g. CENP-B binds the CENP-B-box motif in mammalian centromeric satellites and CENP-B like elements are found even outside of mammals). Motifs are also found in several other fungi with 'regional' centromeres and in certain diatom centromeres. Given that, we expect that, as the quality and density of genome assemblies continues to explode, our predictions could soon be experimentally validated in clades with complex centromeres in other eukaryotic clades. We now also discuss this in the discussion.

We have also attempted to better position the yeast centromere within eukaryotic centromere diversity. The manuscript now contains an overview of the size distribution of eukaryotic centromeres. This overview also highlights that small centromeres are neither rare nor unique to fungi.

3. Centromeres are defined in silico by sequence features, but whether these sequences actually act as centromeres in vivo is not known. To confirm the method of centromere identification it would be necessary to provide some in vivo evidence, for example ChIP-Seq of a kinetochore protein or CENPA.

We used *in vivo* functional assays to demonstrate centromere function. Fig. 2 now includes an additional experiment in which we benchmark our plasmid loss assay by using plasmids without centromere sequences and plasmids with centromere sequences from different species. Plasmids with our predicted sequences are retained significantly more compared to plasmids without a centromere sequence, and outperform plasmids with centromeres from other species:

Plasmid segregation assays are a common way of verifying point centromere sequences, as shown in the updated version of Extended Data Fig. 1 which now also contains a column with information on experimental validation.

4. Similarly, how confident can the authors be in the sequence assemblies, particularly when considering the CDEII lengths? Have the authors carried out directed sequencing to confirm?

We confirmed the centromere sequence of every *J. jinghonensis* centromere (all 16 – key species with a mix of long and short variants) by Sanger sequencing (new Extended Data Fig. 4a):

Our pipeline finds very clear conserved *CDEII* patterns within and across species, without any prior knowledge about how these centromeres should look like. It is therefore also unlikely that these assemblies from many different authors, using different assembly methods, would converge in these very clear evolutionary patterns.

5. Figure 3f. This figure and the conclusion that the variation comes from microhomology mediated recombination is quite difficult to follow. This needs better explanation.

We clarified this in the main text, which now reads:

“Remarkably, the majority of variant centromeres seem to have expanded through a microhomology-mediated mutational mechanism. Aligning longer variants with their most similar short counterparts reveals that the newly inserted sequences are exact copies of short stretches of the original sequence (Fig. 3f).”

We also increased the arrow size in the figures to better highlight the homologous stretches within the sequence.

6. Figure 4c. Can the authors explain why *CDEII* that is 10 bp shorter does not support plasmids stability? Centromeres with a shorter length were identified in Figure 1. How could these work?

The vast majority of species show 1 short and 1 long *CDEII* in cases where they have mixed *CDEII* lengths. The collapsed tree does indeed make it look like there is a +/- symmetry, but this is only true on a genus level. For example: in a genus with a median length of 80 bp, there can be species

with 70 bp and 80 bp variants and *other* species with 80 bp and 90 bp variants. To make this more clear, we added an uncollapsed version of the tree in Fig. 4 as Extended Data Fig. 6. The trend is especially obvious in the *Lachancea* and *Vanderwaltozyma* genus:

In the case of *S. cerevisiae*, most *CDEII* length variants are longer (see variant frequency Fig. 3c). We hypothesize that a species' kinetochore structure will determine whether a -10 or +10 bp variant is preferred, and for *S. cerevisiae*, based on what we see in nature, it seems like longer variants are tolerated, and this is also what we observe in our experiments.

We now also mention this in the main text:

“Similar to what we observed for natural isolates (Fig. 3), *S. cerevisiae* seems to tolerate longer centromere variants better than shorter variants.”

7. The manuscript would be strengthened by further experimental testing of the assertions/hypotheses from the *in silico* data. The authors could, for example, build hybrid centromeres and use exogenous expression to determine whether Cbf1 variants could support different *CDEII* lengths in a heterologous yeast (e.g. in *Saccharomyces cerevisiae*).

Thank you for this great suggestion! We did this experiment, and as mentioned above, we now provide experimental evidence that evolution of a kinetochore protein can indeed alter which centromere variants are preferentially retained:

“To test whether these different Cbf1 variants can indeed alter which centromere variants are preferentially retained, we replaced the endogenous *CBF1* gene in *S. cerevisiae* with different *Jamesozyma* variants, and measured the relative retention rate of plasmids with either short or long *Jamesozyma* centromeres. Remarkably, the *J. spencerorum* Cbf1 protein made long *Jamesozyma* centromeres significantly more efficient than short centromeres (Fig. 4g), suggesting that mutations in this protein might indeed explain why the long variant acquired a selective advantage and reached fixation within the genome.”

Referees' comments:

Referee #1 (Remarks to the Author):

Although the authors provided additional analyses and clarifications in response to previous comments, several of my concerns were only partially addressed. For example, while the inclusion of new analyses in Mucoromycota and additional experimental controls is welcome, some key questions remain. In particular, I remain concerned that, although the results are interesting, they are not particularly surprising in the organisms studied. The manuscript would benefit from more robust information and arguments to support the conclusions, and the extent to which these results can be generalized to systems with more complex centromeres remains unclear. Overall, the study offers intriguing observations and raises interesting hypotheses. However, in its current form, some conclusions appear overstated relative to the data, the generality of the results beyond fungi is questionable, and several claims remain premature without further validation.

We are glad to hear that the reviewer appreciates the new analyses and experimental data that we included in the revised manuscript. We would additionally posit that our manuscript, in its current, revised form, has in fact fully addressed their original concerns.

With regards to the results not appearing to be surprising, we would emphasise that our primary contribution through this work is to provide real evolutionary data and *in vivo* centromere function experiments in support of long-standing hypotheses in the field that have long remained in the realm of pure theory.

In terms of the generality of our conclusions, it is worth pointing out that the *Mucoromycota* are as far from the *Saccharomycetaceae* as humans are from single-celled *Filasteria* (almost a billion years of divergence time, revised upwards thanks to new analyses, just over a week old, in Szanthy et al. *Nature Ecology and Evolution* 2025). The revised manuscript places these robust trends - that we have no reason to believe are in any way restricted to fungi - in the context of eukaryotic centromere diversity, and provides a roadmap for future investigations. We note that Reviewer 3, who was also concerned about the generality of our findings, is now fully satisfied.

Referee #2 (Remarks to the Author):

Helsen et al addressed all my previous comments and their paper is better than their previous submission. The paper is clear and the additions to their current version have made it even more exciting.

We thank the reviewer for their positive and encouraging feedback throughout the revision process.

Few suggestions:

1. In page 6, starting on line 6, we suggest authors to add "centromere LIKELY change one by one".

We added "likely" to this sentence.

2. Figure 3d, we suggest authors show all homozygous chromosome in CBS2271 as it makes it confusing if we are looking at haploids or homozygous diploids.

According to the source of the genome assembly, CBS2271 should be a homozygous diploid strain. We changed this accordingly in the figure.

Referee #3 (Remarks to the Author):

The authors have satisfactorily addressed the points raised by the reviewers. In particular, inclusion of the new data on Mucoromycota has validated the generality of their approach. The authors have also included textual changes comparing the features of point and regional centromeres which is appreciated.

We are happy the reviewer is now satisfied with the generality of our approach.

The authors additionally provide some experimental data that supports the idea that Cbf1 is adapted to Cdell centromere length. This is also appreciated, however, the plasmid retention assay is quite indirect. The conclusion that the phenomenon observed is due to selection "at the kinetochore interface" is premature and at this point rather speculative. Therefore, the title of the manuscript should be toned down to reflect this. On the other hand, the authors have robustly shown that centromeres evolve progressively. The mechanisms that provide the selection to constrain centromere length has not been fully demonstrated in this manuscript.

We now temper our claims regarding the direct impact of the physical interaction between kinetochore and centromere on centromere evolution.

We changed the title of our manuscript to "Progressive coevolution of the yeast centromere and kinetochore".

We further:

- Replaced "indicate" with "are consistent with a model" on page 11 line 37, as suggested.
- Replaced "act as a filter" by "appears to act as a filter" in the abstract.
- Verified that every time we talk about the interaction between the kinetochore and centromere, we specifically use "suggest", and make it clear in the discussion that we "propose a model".

Referee #4 (Remarks to the Author):

I co-reviewed this manuscript with one of the reviewers who provided the listed reports.